# A₂DEPT: Large Language Model–Driven Automated Algorithm Design via Evolutionary Program Trees

Bin Chen [*1]   Shouliang Zhu [*1]   Beidan Liu [2]   Yong Zhao [2]   Tianle Pu [2]   Huichun Li [3]   Zhengqiu Zhu [2]

## Abstract

Designing heuristics for combinatorial optimization problems (COPs) is a fundamental yet challenging task that traditionally requires extensive domain expertise. Recently, Large Language Model (LLM)-based Automated Heuristic Design (AHD) has shown promise in autonomously generating heuristic components with minimal human intervention. However, most existing LLM-based AHD methods enforce fixed algorithmic templates to ensure executability, which confines the search to component-level tuning and limits system-level algorithmic expressiveness. To enable open-ended solver synthesis beyond rigid templates, we propose Automated Algorithm Design via Evolutionary Program Trees (A₂DEPT), which treats LLMs as system-level algorithm architects. A₂DEPT explores the vast program space via a tree-structured evolutionary search with *hybrid selection* and *hierarchical operators*, enabling iterative refinement of complete algorithms. To make open-ended generation practical, we enforce executability with a lightweight program-maintenance loop that performs feedback-driven repair. In experiments, A₂DEPT consistently outperforms representative LLM-based baselines on both standard and highly constrained benchmarks. On the standard benchmarks, it reduces the mean normalized optimality gap by 9.8% relative to the strongest competing AHD baseline.

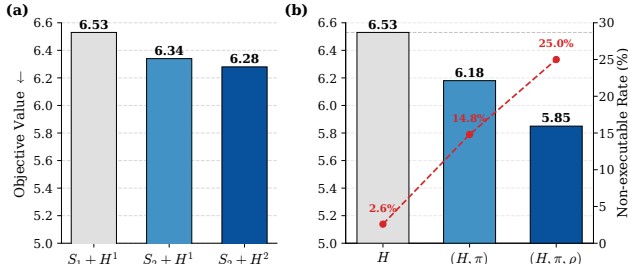

*Figure 1.* **Pilot study on the Traveling Salesman Problem (TSP).** It illustrates (i) framework dependence in template-bound AHD and (ii) the executability challenge when scaling from heuristics to full solver logic. **(a)** $S_1$ is a step-by-step constructive framework, and $S_2$ adds backtracking. $H^1$ and $H^2$ are evolved under $S_1$ and $S_2$. **(b)** We expand the evolution target from a scoring heuristic $H$ to $(H, \pi)$ with backtracking policy $\pi$, and then to $(H, \pi, \rho)$ by adding a utilization mechanism $\rho$. The red dashed curve reports the non-executable rate (generated programs that fail to run).

## 1. Introduction

Combinatorial optimization problems (COPs) (Desale et al., 2015) are ubiquitous in real-world applications, ranging from logistics planning (Bengio et al., 2021) and vehicle routing (Braekers et al., 2016) to chip floorplanning (Mirhoseini et al., 2021) and job scheduling (Chaudhry & Khan, 2016). Due to the combinatorial explosion of the search space, these problems are typically NP-hard (Garey & Johnson, 1979), rendering exact solvers computationally infeasible for large-scale instances. Consequently, constructing high-performance heuristics—which trade optimality for computational efficiency—has become the typical approach to obtain satisfactory solutions under limited time budgets (Talbi, 2009). However, the traditional paradigm of manual heuristic design faces a severe scalability bottleneck. This process is inherently labor-intensive and highly dependent on domain-specific human intuition (Burke et al., 2013). Practitioners must often engage in a tedious trial-and-error cycle to handcraft rules that capture problem structure. Moreover, handcrafted heuristics can be sensitive to problem variants: strategies tuned for one setting may degrade on others (Wolpert & Macready, 2002), requiring substantial re-tuning or redesign.

To address these scalability limitations, Large Language

---

[*]Equal contribution   [1]The Institute of Intelligent Computing, University of Electronic Science and Technology of China, Chengdu 611731, China   [2]College of Systems Engineering, National University of Defense Technology, Changsha 410073, China   [3]Academy of Military Medical Sciences, Beijing 100071, China. Correspondence to: Shouliang Zhu <202522900324@std.uestc.edu.cn>, Zhengqiu Zhu <zhuzhengqiu12@nudt.edu.cn>.

*Proceedings of the 43ʳᵈ International Conference on Machine Learning*, Seoul, South Korea. PMLR 306, 2026. Copyright 2026 by the author(s).

Models (LLMs) have recently enabled a paradigm shift toward Automated Heuristic Design (AHD) (Romera-Paredes et al., 2024). Acting as general-purpose heuristic generators, LLMs can translate high-level intentions into executable code (Ma et al., 2023). This makes it possible to automate substantial parts of the heuristic-development pipeline—from implementing candidate rules to iterating over variants under a user-specified evaluation protocol—thereby reducing reliance on repeated manual re-engineering.

Recent studies have further integrated LLMs with Evolutionary Computation (EC) to advance AHD. For instance, FunSearch (Romera-Paredes et al., 2024) adopts an LLM-as-mutation-operator paradigm to evolve priority functions. Subsequent methods further improve generation quality: EoH (Liu et al., 2024a) evolves natural-language thoughts and code jointly, while ReEvo (Ye et al., 2024) leverages reflective verbal gradients for self-improvement. Most recently, MCTS-AHD (Zheng et al., 2025) introduces Monte Carlo Tree Search to mitigate premature convergence and better balance exploration and exploitation.

Despite these advances, a critical limitation persists: most LLM-based AHD methods design key heuristic functions within predefined solver frameworks. Advanced search strategies are therefore applied only to optimize isolated functional slots. As a result, performance becomes tightly coupled to the chosen framework. Figure 1(a) illustrates the bottleneck: heuristics evolved under one constructive framework can fail to transfer under another, and the relative advantage of structural choices can be problem-dependent. Consequently, template-constrained AHD inherits a hidden yet consequential design decision—selecting an appropriate solver backbone—that is difficult to resolve a priori, and can cap performance even when the heuristic component itself is well optimized.

These limitations motivate a transition from template-bound AHD to open-ended Automated Algorithm Design (AAD), where the search target expands from heuristic components to complete solver programs, as illustrated in Figure 2(a). By enabling redesign of the full algorithmic workflow, AAD unlocks a much richer design space, but it also amplifies generation difficulty. As Figure 1(b) suggests, enlarging the evolution target from a scoring heuristic to broader solver control can improve objective values, yet it increases the non-executable rate due to the growing complexity of free-form code generation. This transition introduces three fundamental challenges. First, **Executability Instability**: complete solvers involve long, interdependent code with fragile interfaces and dependencies, making candidates prone to compilation/runtime failures. Second, **Search Space Explosion**: AAD must explore an effectively unbounded program space where valid, high-performing algorithms are sparse. Third, **Credit Assignment Opacity**: large coupled edits

change multiple behaviors at once, making it difficult to attribute performance differences to specific modifications and weakening search guidance.

In this paper, we propose **A$_2$DEPT** (**A**utomated **A**lgorithm **D**esign via **E**volutionary **P**rogram **T**rees), a framework for open-ended AAD that evolves complete solver programs. A$_2$DEPT performs tree-structured program search with hybrid selection, combining SA-style parent–child acceptance with Boltzmann sampling to preserve diverse trajectories. It further evolves programs with hierarchical operators (micro-tuning, macro-mutation, semantic crossover) under adaptive scheduling. To keep open-ended evolution runnable, we couple the search with a lightweight maintenance loop that repairs missing dependencies via a dependency graph and prunes unreachable code.

Our main contributions are concluded as follows:

- **Problem framing.** We identify the framework bottleneck of LLM-based AHD and motivate open-ended AAD as program-space search over executable solvers.

- **A$_2$DEPT framework.** We propose A$_2$DEPT, a tree-structured evolutionary program search framework that uses LLMs as program-level variation operators to evolve complete solvers.

- **Mechanisms for reliable long-horizon evolution.** We introduce (i) a program-maintenance loop for dependency repair and executability enforcement, (ii) a tree-structured hybrid selection scheme that preserves diverse trajectories under a fixed budget, and (iii) hierarchical operators with adaptive scheduling to enable localized edits and improved credit assignment.

- **Empirical validation.** We evaluate A$_2$DEPT on a diverse suite of NP-hard combinatorial optimization benchmarks (standard and high-constraint) and show consistent gains over representative LLM-based AHD baselines. On the standard benchmarks, A$_2$DEPT reduces the mean normalized optimality gap by **9.8%** relative to the strongest competing AHD baseline, where each per-task gap is defined as the relative percentage deviation from the optimum before averaging.

**Conflict of Interest Disclosure.** The authors declare no financial conflicts of interest related to this work.

## 2. Background and Related Work

### 2.1. Heuristics for Combinatorial Optimization

Combinatorial optimization problems such as TSP (Rosenkrantz et al., 1974) and Job Shop Scheduling (JSP) (Jain & Meeran, 1999) are typically NP-hard, making exact

solvers impractical at scale. Heuristics therefore serve as the workhorse in practice, broadly falling into (i) *constructive* rules that build feasible solutions incrementally (Campbell & Savelsbergh, 2004; Paessens, 1988) and (ii) *improvement* methods that iteratively refine incumbents via local search or metaheuristics (Dorigo et al., 2007; Katoch et al., 2021), e.g., simulated annealing (Rutenbar, 2002) and tabu search (Al-Sultan, 1995). However, hand-crafted heuristics are often brittle: they are tuned to specific instance distributions and can degrade under shifting constraints or operating conditions (Hooker, 1995).

## 2.2. Automated Heuristic Design

Automated Heuristic Design (Liu et al., 2024a), often studied under the hyper-heuristics paradigm, aims to automate the construction of optimization heuristics. Early AHD work was dominated by Genetic Programming (GP) (Langdon & Poli, 2013; Koza, 1994), which evolves heuristic programs via crossover and mutation but can suffer from instability (e.g., invalid programs and code bloat) (O'Neill et al., 2010). More recently, LLMs have enabled semantically informed program generation and editing for AHD (Liu et al., 2025; van Stein & Bäck, 2024), exemplified by FunSearch (Romera-Paredes et al., 2024), EoH (Liu et al., 2024a), ReEvo (Ye et al., 2024), and tree-based variants such as MCTS-AHD (Zheng et al., 2025). However, most LLM-based AHD methods remain template-bound: they optimize a small set of heuristic components (e.g., scoring/priority functions) within predefined algorithmic templates, while keeping the surrounding control flow and solver logic fixed. This structural constraint limits their ability to discover system-level algorithmic innovations required by complex optimization problems. A$_2$DEPT performs semantically informed edits over complete solver programs with explicit executability checks, enabling system-level redesign rather than component-level tuning.

## 2.3. LLMs for Program Synthesis and Optimization

Recent advances in LLMs have catalyzed progress in program synthesis and code-level optimization (Xi et al., 2025; Lu et al., 2023; Shypula et al., 2023; Liventsev et al., 2023). A dominant paradigm integrates LLM-based generation with evaluation-driven feedback (Yang et al., 2023): instead of one-shot code synthesis, systems interleave generation with execution, verification, or performance assessment to iteratively refine programs (Wang et al., 2023; Wu et al., 2024). Such generation–evaluation loops have been explored across debugging (Chen et al., 2023), prompt optimization (Zhou et al., 2022), reward design (Ma et al., 2023), robotics (Lehman et al., 2023), and broader search/evolution-based optimization settings. However, program-space search remains challenging due to the fragility of code edits and the sheer size of the search space (Chen, 2021; Austin

et al., 2021). In contrast to prior work that typically optimizes programs for correctness or local efficiency under fixed specifications/tests, A$_2$DEPT targets open-ended algorithmic performance and evolves complete solver logic via benchmark-driven feedback.

## 3. Methodology

### 3.1. Workflow Overview

As illustrated in Figure 2, A$_2$DEPT casts AAD as an evolutionary search over a discrete program space. It maintains a global search tree $\mathcal{T}$ (Zheng et al., 2025) of *executable programs*, where each node stores the program, its score, and its generation metadata. Unlike population-based methods, $\mathcal{T}$ retains all evaluated programs, enabling structured exploration and reuse of prior trajectories. Formally, each node is defined as a tuple $n = (P, S, \mathbf{h}, \omega)$, where $P$ is an executable program, $S$ is its score on $\mathcal{D}$, $\mathbf{h}$ records the parent/operator history, and $\omega$ stores local operator weights.

A$_2$DEPT decomposes long-horizon solver synthesis into a sequence of localized, verifiable program edits guided by empirical feedback. As shown in Figure 2(b), it expands a *frontier* in batches: each iteration selects a *set* of parent nodes via hybrid selection and applies adaptively scheduled operators to generate multiple independent offspring. To prevent failures from propagating across these expansions, the program-maintenance pipeline in Figure 2(c) repairs broken dependencies and prunes unreachable code before evaluation, and the scores are written back into $\mathcal{T}$ to guide subsequent selection.

The workflow proceeds in two phases. First, an LLM-based initialization produces a diverse set of executable root programs (see Appendix C.1). Then, A$_2$DEPT enters an iterative loop: it selects parent nodes, applies hierarchical mutation and recombination to generate new code, repairs broken dependencies via the program maintenance mechanism, evaluates feasible programs, and inserts them back into $\mathcal{T}$ until the computational budget is exhausted. The specific mechanisms for selection, generation, and governance are detailed in the following subsections.

**Design rationale.** The components of A$_2$DEPT are not an interchangeable assembly; each is necessitated by the consequences of the previous one, forming a self-reinforcing loop tailored to open-ended AAD. Tree-structured search with SA-style parent–child acceptance is a natural fit because AAD primarily requires *describing parent–child refinement relations* rather than maintaining population-level dynamics; SA satisfies detailed balance and provides a well-characterized exploration–exploitation trade-off. However, SA's rejection rate is inherently high in open-ended AAD, since many children carry structural changes whose bene-

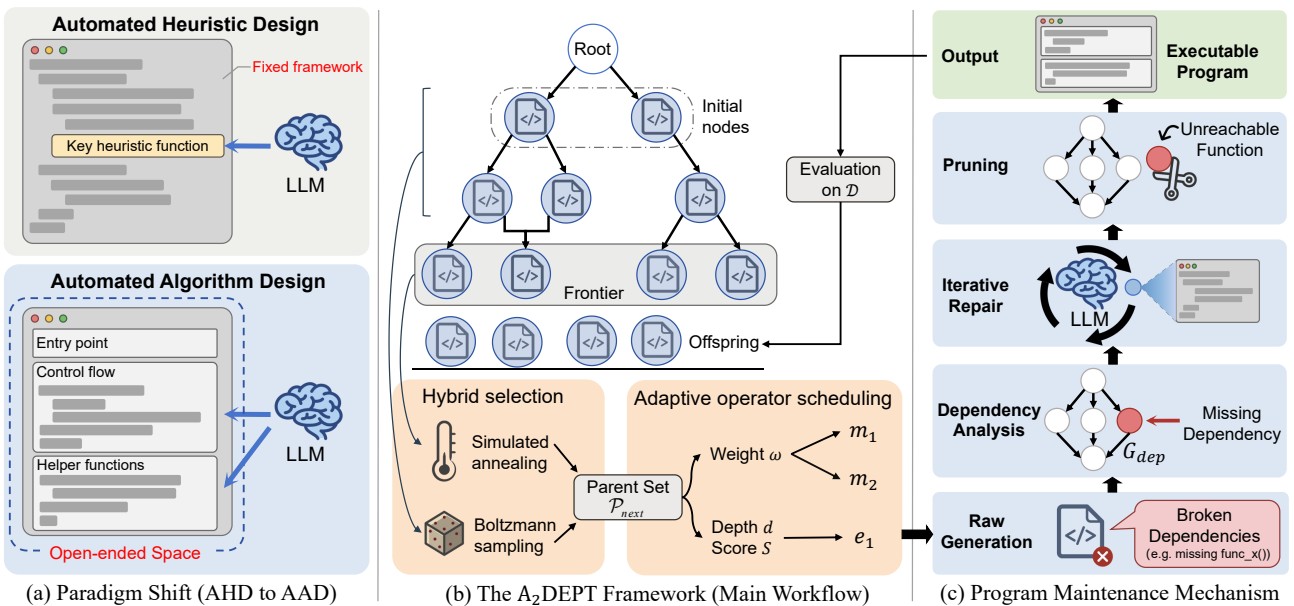

*Figure 2.* **Overview of the A₂DEPT framework.** **(a)** Paradigm shift from template-bound AHD, where the LLM fills heuristic slots within a fixed solver framework, to open-ended AAD, where the LLM synthesizes complete solver programs. **(b)** A₂DEPT maintains a global search tree and expands a frontier in a batched manner via hybrid selection and adaptive operator scheduling. **(c)** A program maintenance pipeline enforces executability through dependency analysis, iterative repair, and pruning, yielding executable programs for evaluation on $\mathcal{D}$.

fits are not immediately reflected in the score; this shrinks the parent frontier and motivates Boltzmann supplementary sampling over $\mathcal{T}$ to recover temporarily underperforming but structurally promising branches. The resulting diversity provides the stepping stones that macro-mutation needs to achieve paradigm-level restructuring, whose free-form code in turn *necessitates* the dependency-repair loop to preserve executability. Finally, evaluation feedback drives the adaptive scheduler (Eq. 5–6), which learns which operator granularity is productive at each node, shaping the next cycle. The ablation in Table 4 confirms that removing any one component breaks the loop and causes clear degradation on at least one task.

**3.2. Program Maintenance Mechanism**

Algorithms in AAD are inherently complex, involving multiple interdependent functions, global variables, and library calls. Evolving such systems as flat text sequences is fragile, as a single syntax error or unresolved reference can invalidate the entire *program*. To address this issue, A₂DEPT introduces a program maintenance mechanism that improves executability during evolution.

3.2.1. STRUCTURED PROGRAM REPRESENTATION

Instead of evolving raw text, A₂DEPT represents each solver as a function-level modular program.

Formally, a program is written as $P = (\mathcal{C}_{pre}, \mathcal{F}_{funcs})$. The

*Preface* $\mathcal{C}_{pre}$ contains global imports/configurations (and constants) and is kept fixed to stabilize the execution environment. The *Function Registry* $\mathcal{F}_{funcs}$ is a set of modular functions, where each function $f$ is represented by its signature and executable body.

To regulate evolution, A₂DEPT applies a role-aware parsing that partitions $\mathcal{F}_{funcs}$ into *Immutable Definitions* $\mathcal{F}_{imm}$ and *Mutable Strategies* $\mathcal{F}_{mut}$. Operators act only on $\mathcal{F}_{mut}$ (Section 3.4), which prevents accidental corruption of foundational utilities in $\mathcal{F}_{imm}$ and makes micro-tuning edits interface-preserving.

3.2.2. DEPENDENCY REPAIR AND CLOSURE

Structural mutations may introduce missing dependencies when generated code calls undefined functions. A₂DEPT resolves this via an automated dependency-repair mechanism.

Given a program $P$, we build a directed call graph $G_{\text{dep}} = (V, E)$, where $V$ is the set of function names appearing in $P$ and $(u, v) \in E$ denotes that $u$ calls $v$. Let $\text{Def}(P)$ be the set of function names defined in $P$, and let $\mathcal{B}$ denote language built-ins. We identify unresolved symbols as

$$\mathcal{U} = \{\, v \in V \mid \exists u \in V : (u, v) \in E \land v \notin (\text{Def}(P) \cup \mathcal{B}) \,\}.$$
(1)

If $\mathcal{U} \neq \emptyset$, A₂DEPT iteratively prompts the LLM to implement the missing functions using call-site context (signatures and usage) until $\mathcal{U} = \emptyset$, yielding a dependency-closed program. The repair loop is guaranteed to terminate: it halts

either when all missing dependencies are resolved ($\mathcal{U} = \emptyset$) or when the global LLM call budget is exhausted. In the latter case, the program that remains incomplete is assigned a score of zero and does not block subsequent search iterations (Algorithm 1, Lines 19–26 in Appendix B).

Finally, we perform reachability analysis from the program entry function and prune functions that are unreachable, preventing dead logic from accumulating over iterations.

### 3.3. Tree-Structured Hybrid Search Strategy

To enable efficient exploration of the vast and discrete AAD space, A$_2$DEPT maintains a global search tree $\mathcal{T}$. At each iteration, the goal is to select a set of valid parent nodes $\mathcal{P}_{next}$ from the current frontier to generate the next batch of candidate programs. Because $\mathcal{T}$ primarily serves as a memory of parent–offspring transformation trajectories rather than a rollout-intensive planning structure, we adopt a lightweight hybrid rule: simulated-annealing acceptance explicitly reflects the parent–child refinement relation with minimal overhead, and Boltzmann sampling over $\mathcal{T}$ supplements the remaining budget to reach a fixed parent set size $K$ while avoiding greedy top-$k$ collapse.

#### 3.3.1. SIMULATED ANNEALING-BASED PRIMARY SELECTION

The primary selection stage determines which nodes from the most recent expansion of the search tree are selected for further evolution. We employ a simulated annealing strategy (Nourani & Andresen, 1998) based on the Metropolis criterion to filter the current frontier. For each newly generated node $n_c$, its performance is compared against that of its parent $n_p$ to determine whether it should be retained for the subsequent evolutionary cycle.

Assuming that the scoring function $S(n)$ is to be maximized, the selection probability $P_{\text{select}}$ is defined as:

$$P_{\text{select}}(n_c, n_p) = \begin{cases} 1 & \text{if } S(n_c) \geq S(n_p), \\ \exp\left(\frac{S(n_c) - S(n_p)}{T_t}\right) & \text{otherwise,} \end{cases} \quad (2)$$

where $T_t$ denotes the system temperature at iteration $t$. Nodes accepted under this criterion are added to $\mathcal{P}_{next}$ as seeds for subsequent mutations.

*Dynamic Re-annealing.* To mitigate search stagnation, A$_2$DEPT monitors the global best solution $P^*$. If $P^*$ remains unchanged for a predefined number of generations $N_{\text{stall}}$, the temperature $T_t$ is increased to restore exploration capability (Ingber, 1989). The temperature update rule is given by:

$$T_{t+1} = \begin{cases} T_t + \Delta T & \text{if } \Delta t_{\text{stall}} \geq N_{\text{stall}}, \\ \alpha T_t & \text{otherwise,} \end{cases} \quad (3)$$

where $\alpha \in (0, 1)$ is a decay factor and $\Delta T$ denotes the re-annealing increment. Upon triggering re-annealing, the stagnation counter $\Delta t_{\text{stall}}$ is reset to zero, allowing the standard temperature decay schedule to resume.

#### 3.3.2. BOLTZMANN-BASED SUPPLEMENTARY SELECTION

Since the simulated annealing stage may reject many candidates, the resulting parent set may contain fewer than $K$ nodes (i.e., $|\mathcal{P}_{next}| < K$). To maintain constant parallel throughput, A$_2$DEPT supplements the parent set by sampling additional nodes from the global search tree $\mathcal{T}$.

Specifically, $K - |\mathcal{P}_{next}|$ nodes are selected according to Boltzmann selection (Goldberg, 1990), where the probability of choosing a node $n_i$ is defined as:

$$P_{\text{supp}}(n_i) = \frac{\exp\left((S(n_i) - S_{\max})/T_t\right)}{\sum_{n_j \in \mathcal{T}} \exp\left((S(n_j) - S_{\max})/T_t\right)}, \quad (4)$$

with $S_{\max}$ denoting the maximum score observed in the search history.

Unlike greedy strategies that exclusively favor top-performing candidates, this probabilistic selection assigns non-zero sampling probabilities to suboptimal nodes.

### 3.4. Hierarchical Operators with Adaptive Scheduling

A challenge in AAD is coarse-grained credit assignment: when a program is modified, it is non-trivial to attribute an observed performance change to specific structural edits. To address this, A$_2$DEPT decomposes evolution into hierarchical granularities and uses an adaptive scheduler to dynamically allocate search resources.

#### 3.4.1. HIERARCHICAL OPERATOR DEFINITION

We define a tiered operator set $\mathcal{O} = \{m_1, m_2, e_1\}$ that acts at three levels of program transformation: **micro-level refinement**, **macro-level workflow restructuring**, and **semantic recombination** via crossover.

*Micro-tuning ($m_1$):* Applies localized edits to the mutable strategy functions $\mathcal{F}_{mut}$ (Section 3.2), refining logic and parameters while preserving interfaces.

*Macro-mutation ($m_2$):* Rewrites the solver workflow by reconstructing the entry-point function, allowing changes to control flow and function signatures; newly introduced dependencies are resolved by the maintenance loop.

*Semantic crossover ($e_1$):* Synthesizes a hybrid program from two parents ($n_a, n_b$) by integrating complementary mechanisms from their code and performance feedback, with missing components repaired automatically.

### 3.4.2. ADAPTIVE SCHEDULING LOGIC

Instead of using fixed operator probabilities, A$_2$DEPT maintains a node-specific weight vector $\boldsymbol{\omega}_n = (\omega_{m_1}, \omega_{m_2})$ for each node $n$. At node $n$, the probability of selecting a mutation operator $op \in \{m_1, m_2\}$ is given by a softmax scheduler:

$$P(op \mid n) = \frac{\exp(\omega_{op}/\tau)}{\sum_{k \in \{m_1, m_2\}} \exp(\omega_k/\tau)}, \qquad (5)$$

where $\tau > 0$ is a temperature parameter regulating the exploration-exploitation trade-off. The crossover operator $e_1$ is invoked conditionally and is therefore excluded from the node-local scheduling distribution.

After generating a child node $n_{\text{child}}$ from parent $n_{\text{parent}}$ using operator $op$, A$_2$DEPT computes a normalized performance change $\bar{r} = \left(S(n_{\text{child}}) - S(n_{\text{parent}})\right)/|S(n_{\text{parent}})|$. The corresponding operator weight is then updated to reinforce beneficial modifications:

$$\omega_{op}^{t+1} = \begin{cases} \omega_{op}^t + \bar{r} & \text{if } \bar{r} \geq 0, \\ \omega_{op}^t - \lambda|\bar{r}| & \text{if } \bar{r} < 0, \end{cases} \qquad (6)$$

where $\lambda > 0$ penalizes unsuccessful mutations. The child node $n_{\text{child}}$ inherits the updated vector $\boldsymbol{\omega}_{n_{\text{child}}} \leftarrow \boldsymbol{\omega}_{n_{\text{parent}}}^{t+1}$ as a *warm start* rather than a fixed preference: because the softmax scheduler (Eq. 5) with temperature $\tau$ always assigns non-zero probability to both operators, an inappropriate inherited bias (e.g., after a macro-mutation that substantially alters program structure) is corrected by subsequent feedback within a few iterations, as Eq. 6 penalizes the underperforming operator and reinforces the productive one. Overall, this feedback-driven scheduler adaptively shifts search effort between micro-tuning and macro-mutation as the search progresses.

### 3.4.3. DIVERSITY-AWARE PARTNER SELECTION

We select crossover partners by balancing performance and trajectory diversity. For a primary parent $n_a$, we measure diversity with the depth of the least common ancestor (LCA) in the search tree: smaller $D(\text{LCA}(n_a, n_b))$ indicates earlier divergence. We sample $n_b$ with

$$P_{\text{cross}}(n_a, n_b) \propto \frac{S(n_m) - S_{\min}}{S_{\max} - S_{\min}} \left(1 - \frac{D(\text{LCA}(n_a, n_b))}{D_{\max}}\right), \qquad (7)$$

where $n_m = \arg\max\{S(n_a), S(n_b)\}$, $D(\cdot)$ is node depth, and $D_{\max}$ is the current maximum depth.

## 4. Experiments

In this section, we evaluate A$_2$DEPT on six combinatorial optimization problems, reporting solution quality on standard benchmarks, robustness across alternative design paradigms, and feasibility under stringent constraints; we further conduct ablation studies, parameter sensitivity analyses and a case study. Additional analyses are deferred to Appendix E, including robustness to the LLM backbone (Appendix E.1), comparison with LLM-as-Optimizer (Appendix E.2), convergence behavior at different search budgets (Appendix E.3), LLM scaling and equalized token-budget studies (Appendix E.4), hyperparameter sensitivity (Appendix E.5), wall-clock budget sensitivity (Appendix E.6), the source of the GLS advantage on FJSP (Appendix E.7), out-of-distribution generalization (Appendix E.8), and resource consumption (Appendix E.9).

**Benchmarks.** We evaluate A$_2$DEPT on a diverse set of NP-hard CO problems. For standard performance evaluation, we consider four representative benchmarks from FrontierCO (Feng et al., 2025): Maximum Independent Set (MIS), Capacitated Vehicle Routing Problem (CVRP), Capacitated Facility Location Problem (CFLP), and Flexible Job-Shop Scheduling Problem (FJSP). To further assess robustness in highly constrained settings, we additionally include the Capacitated Electric Vehicle Routing Problem with Time Windows (CEVRPTW) and the Multi-Mode Resource Constrained Project Scheduling Problem (MRCPSP). Detailed problem definitions are provided in Appendix A.4.

**Baselines.** We compare A$_2$DEPT against three categories of baselines. (1) classical solvers: We include the commercial solver Gurobi (Gurobi Optimization, LLC, 2025) as a reference for optimal or near-optimal solutions, together with a classical heuristic baseline based on Iterated Local Search (ILS) (Lourenço et al., 2003). (2) learning-based methods: We evaluate representative neural solvers tailored to each problem, including SDDS (Sanokowski et al., 2025) for MIS, DeepACO (Ye et al., 2023) for CVRP, GCNN (Gasse et al., 2019) for CFLP, and MPGN (Lei et al., 2022) for FJSP. (3) LLM-based AHD methods: We compare against recent LLM-driven heuristic design frameworks, namely FunSearch (Romera-Paredes et al., 2024), EoH (Liu et al., 2024a), ReEvo (Ye et al., 2024), and MCTS-AHD (Zheng et al., 2025). For standard benchmarks and high-constraint evaluations, we use a unified step-by-step constructive framework (Asani et al., 2023), which incrementally builds feasible solutions with minimal domain-specific engineering and thus emphasizes the LLM's heuristic design capability. For the generalization study (Section 4.1.2), we instead evaluate under the Guided Local Search(GLS) and AAD paradigms. A detailed description on these general frameworks is provided in Appendix A.

**Settings.** A$_2$DEPT is implemented in Python and executed on a personal computer equipped with an Intel Core Ultra 9 285K CPU and 96 GB of system memory. For all LLM-based methods, we fix the search budget to $N{=}500$

LLM calls. Baseline methods are configured following the hyperparameters reported in their original papers. For fairness, each generated solver program is given a wall-clock time limit of 120 seconds to solve the entire evaluation dataset $\mathcal{D}_{eval}$ (Appendix D.2). We run each method three times on standard benchmarks and twenty times on high-constraint benchmarks, reporting averaged results (and standard deviation when applicable). Within A$_2$DEPT, we set the parent budget to $K=5$ per expansion step. Simulated annealing is initialized with $T_0=1.0$ and decay factor $\alpha=0.95$, with re-annealing triggered after $N_{stall}=3$ consecutive generations without improvement using increment $\Delta T=0.2$. We synchronize the scheduler temperature $\tau_t$ with the annealing temperature $T_t$ and set the penalty coefficient to $\lambda=0.8$.

## 4.1. Main Results

### 4.1.1. EXPERIMENTS ON STANDARD BENCHMARKS

As shown in Table 1, A$_2$DEPT outperforms all LLM-based AHD baselines on every benchmark under both DeepSeek and Gemini, under the same step-by-step constructive template. Beyond winning on all tasks, A$_2$DEPT delivers sizable and consistent gap reductions relative to the best competing AHD method (for instance, from 17.42% to 8.79% on CVRP and from 11.35% to 4.20% on MIS under DeepSeek), indicating that solver-level edits translate into robust quality gains. Averaging the per-task normalized Gap(%) values in Table 1 (each already expressed as a percentage deviation from the optimum, so cross-task averaging is on a common scale), A$_2$DEPT reduces the mean gap by $9.8\%$ relative to the best competing AHD method under DeepSeek. On CVRP and FJSP, the gains are smaller, indicating that domain-specific inductive biases and mature hand-engineered operators narrow the headroom available to generic program-space evolution. Notably, A$_2$DEPT is competitive with strong non-LLM references: it matches Gurobi on MIS while remaining clearly better than all LLM baselines, whereas on CVRP the specialized neural method remains ahead, highlighting that learned priors can still dominate when problem structure is well captured by training. Overall, the table shows that A$_2$DEPT consistently converts open-ended program exploration into measurable quality gains, with the largest benefits arising in domains where solver-level redesign matters most.

### 4.1.2. GENERALIZATION ACROSS DESIGN FRAMEWORKS

To ensure that the observed AAD–AHD gap is not specific to the step-by-step constructive template, we additionally evaluate the AHD baselines under GLS framework, where the LLM designs knowledge-based guidance matrices while the local-search operators are fixed (Appendix A.3). We also evaluate EoH and ReEvo in AAD.

As shown in Table 2, FJSP is an outlier where GLS outperforms all AAD-style methods when provided with expert-designed neighborhood operators, suggesting that a local-search inductive bias can dominate benefits of open-ended solver synthesis. This also suggests AHD can be competitive when the framework supplies suitable operator set and the LLM only needs to learn effective guidance. To pinpoint the source of the GLS advantage on FJSP, we additionally compare the LLM-designed guidance matrix against a random one under the same fixed operators (Appendix 14): the random matrix loses only a modest amount on FJSP but causes a sharp drop on CFLP. This contrast indicates that the FJSP advantage stems largely from the strong inductive bias of expert-designed neighborhood operators rather than from LLM-designed knowledge, making template-bound AHD and open-ended AAD *complementary* rather than strictly ordered. In contrast, A$_2$DEPT targets a more general setting that evolves solver logic directly in program space without domain-specific operators, and it remains consistently strong on CFLP, CVRP, and MIS. Meanwhile, simply moving EoH and ReEvo from AHD to AAD does not close the gap to A$_2$DEPT, suggesting that extra freedom must be paired with mechanisms that preserve long-range logical consistency to be useful.

### 4.1.3. EXPERIMENTS ON HIGH-CONSTRAINT CO

High-constraint combinatorial optimization problems present significant challenges due to fragmented search spaces and stringent validation requirements, making the identification of feasible solutions particularly difficult. To evaluate A$_2$DEPT's capacity to address these challenges, we conduct experiments on two representative problems: CEVRPTW and MRCPSP. For CEVRPTW, we use the ES-OGU benchmark (Aslan et al., 2025). For MRCPSP, we use instances from the PSPLIB library (Sprecher & Kolisch, 1996). Detailed dataset information is provided in Appendix D.1. We report two metrics: the Infeasibility Rate (IR), which measures the percentage of test instances where the algorithm fails to generate a valid solution, and the Relative Gap, which quantifies the deviation from the best solution achieved across all successful runs. The results in Table 3 highlight A$_2$DEPT's performance in high-constraint environments. On the CEVRPTW task, A$_2$DEPT achieves the **lowest IR and the best performance** in terms of optimality gap. For the MRCPSP problem, A$_2$DEPT outperforms all other methods in terms of solution feasibility. While no method achieves a valid solution for all instances due to the problem's complexity, A$_2$DEPT demonstrates robustness with a **substantially lower IR**.

## 4.2. Ablation Studies

To validate the necessity of A$_2$DEPT's components, we construct several ablated variants by disabling one mech-

*Table 1.* **Main Results on Standard Benchmarks.** Comparison of A$_2$DEPT against baselines under different LLM backbones. Obj. denotes the objective value (lower ↓ or higher ↑ is better), and Gap reports the relative gap (%) to the Optimal solution. Standard deviations are shown for Gap to emphasize robustness under repeated runs. For LLM-based methods, we additionally report the standard deviation over independent runs. Bold: global best results among non-oracle methods; Shaded: best performance within each LLM group.

| Method | CFLP | | CVRP | | FJSP | | MIS | |
|---|---|---|---|---|---|---|---|---|
| | Obj. ($10^5$)↓ | Gap (%) ± std ↓ | Obj. ($10^5$)↓ | Gap (%) ± std ↓ | Obj. ($10^4$)↓ | Gap (%) ± std ↓ | Obj. ($10^3$)↑ | Gap (%) ± std ↓ |
| ***Baselines*** | | | | | | | | |
| Optimal | 6.01 | – | 0.99 | – | 1.25 | – | 2.86 | – |
| ILS | 8.36 | 39.04 | 1.18 | 19.48 | 1.50 | 19.78 | 2.31 | 19.11 |
| Neural* | 6.51 | 8.26 | **1.05** | **6.31** | 1.46 | 16.55 | 2.45 | 14.26 |
| Gurobi | **6.01** | **0.00** | 1.06 | 6.75 | **1.34** | **7.22** | 2.74 | 4.28 |
| ***LLM: DeepSeek v3.2 (Non-Think Mode)*** | | | | | | | | |
| FunSearch | 7.34 | 22.13±3.03 | 1.23 | 24.13±9.59 | 1.60 | 27.82±3.98 | 2.35 | 17.89±2.93 |
| EoH | 7.12 | 18.48±4.73 | 1.16 | 17.42±1.55 | 1.60 | 28.08±1.62 | 2.54 | 11.35±6.18 |
| ReEvo | 7.01 | 16.56±6.55 | 1.20 | 18.04±4.65 | 1.60 | 28.24±2.71 | 2.44 | 14.70±3.78 |
| MCTS-AHD | 6.86 | 14.09±2.23 | 1.19 | 19.97±3.88 | 1.58 | 26.55±2.13 | 2.40 | 16.16±5.51 |
| A$_2$DEPT (Ours) | 6.70 | **11.46±3.70** | 1.08 | **8.79±2.63** | 1.49 | **18.95±4.15** | **2.74** | **4.20±2.07** |
| ***LLM: Gemini 2.5 Flash*** | | | | | | | | |
| FunSearch | 7.45 | 23.94±1.69 | 1.24 | 25.13±3.96 | 1.62 | 29.31±5.15 | 2.37 | 17.12±6.28 |
| EoH | 7.08 | 17.82±8.04 | 1.14 | 14.72±0.79 | 1.62 | 29.45±3.72 | 2.39 | 16.47±4.27 |
| ReEvo | 6.98 | 16.15±6.75 | 1.20 | 17.47±1.38 | 1.62 | 29.21±4.09 | 2.40 | 16.15±0.83 |
| MCTS-AHD | 7.26 | 20.87±2.57 | 1.14 | 15.28±3.73 | 1.63 | 30.16±1.45 | 2.40 | 16.14±5.90 |
| A$_2$DEPT (Ours) | 6.87 | **14.24±1.25** | 1.11 | **12.49±2.73** | 1.53 | **22.71±5.91** | 2.65 | **7.43±2.92** |

Note: "Neural*" denotes learning-based methods.

*Table 2.* **Generalization across Paradigms.** Comparison of optimality gaps (%) under different algorithm design frameworks on standard benchmarks. Values are reported as mean gap ± standard deviation. All LLM-based methods use DeepSeek v3.2 (Non-Think Mode). Lower is better ↓. Bold: best performance in each column.

| Method | CFLP | CVRP | FJSP | MIS |
|---|---|---|---|---|
| A$_2$DEPT (Ours) | **11.46±3.70** | **8.79±2.63** | 18.95±4.15 | **4.20±2.07** |
| ***Framework: GLS*** | | | | |
| FunSearch | 20.92±4.52 | 12.62±2.11 | 14.71±1.82 | 17.88±0.41 |
| EoH | 17.39±1.63 | 12.68±4.63 | 11.71±1.09 | 15.33±3.41 |
| ReEvo | 14.39±1.85 | 12.32±3.65 | 13.77±0.52 | 17.79±2.90 |
| MCTS-AHD | 15.87±3.55 | 12.09±2.87 | **10.76±1.06** | 10.49±1.74 |
| ***Framework: AAD*** | | | | |
| EoH | 16.51±8.27 | 10.38±3.19 | 17.47±4.66 | 8.31±3.99 |
| ReEvo | 13.79±6.23 | 9.72±4.65 | 19.67±7.73 | 7.85±2.51 |

*Table 3.* **Results on High-Constraint Problems.** Comparison of IR and Optimality Gap. Gap is relative to the known optimum; "–" indicates the optimum is unavailable. Bold: best result.

| Method | CEVRPTW | | MRCPSP | |
|---|---|---|---|---|
| | Gap (%) ↓ | IR (%) ↓ | Gap (%) ↓ | IR (%) ↓ |
| ***LLM: DeepSeek v3.2 (Non-Think Mode)*** | | | | |
| FunSearch | 6.81 | 15.25 | – | 29.64 |
| EoH | 3.52 | 9.87 | – | 25.68 |
| ReEvo | 4.89 | 8.67 | – | 22.75 |
| MCTS-AHD | 3.75 | 8.79 | – | 24.85 |
| A$_2$DEPT (Ours) | **0.00** | **2.56** | – | **15.89** |

anism from the full pipeline, with all other settings fixed. Part A of Table 4 shows that each ablation degrades performance on at least one task relative to full A$_2$DEPT. In particular, disabling adaptive scheduling increases the gap on both tasks, suggesting that feedback-driven allocation is important for balancing local refinement and structural edits. Removing Boltzmann supplementation further degrades performance, supporting the role of diversity-preserving sampling from the global search tree. Replacing hybrid SA+Boltzmann selection with uniform random parent se-

lection also worsens both benchmarks, confirming the importance of performance-aware selection pressure. Finally, constraining the method to a fixed-template AHD setting yields the largest gaps, highlighting the benefit of open-ended program-level solver evolution. We also analyze sensitivity to the parent set size $k$, which controls the batch expansion budget. Part B of Table 4 shows that overly small $k$ limits diversity, whereas overly large $k$ dilutes the fixed sampling budget; overall, $k=5$ offers the best trade-off (additional results are reported in Appendix E.5).

*Table 4.* **Ablation Studies and Parameter Sensitivity.** Comparison of Optimality Gap (%) on CVRP and FJSP. Bold: best result.

| Configuration | Gap (%) | |
|---|---|---|
| | CVRP | FJSP |
| ***A. Core Component Ablation*** | | |
| Variant-I (Fixed-template) | 16.15 | 26.30 |
| Variant-II (w/o Boltzmann) | 12.73 | 22.11 |
| Variant-III (w/o Adaptive) | 9.59 | 19.78 |
| Variant-IV (Random selection) | 10.32 | 20.02 |
| A$_2$DEPT (Ours) | **8.79** | **18.95** |
| ***B. Parameter Sensitivity (Parent Set Size $k$)*** | | |
| A$_2$DEPT ($k = 3$) | 9.22 | 21.69 |
| A$_2$DEPT ($k = 5$) | 8.79 | **18.95** |
| A$_2$DEPT ($k = 7$) | **8.66** | 23.81 |

### 4.3. Case Study

We trace the evolutionary lineage of the best MIS solver to illustrate how A$_2$DEPT's mechanisms interact (Appendix F). A branch rejected by primary selection is recovered by Boltzmann supplementation, preserving a diverse stepping stone. Along this branch, hierarchical operators enable a jump from a constructive greedy heuristic to an ILS-style solver with perturbation and multi-operator local search. Program maintenance makes this expansion practical by repairing dependencies and enforcing executability. Subsequent iterations refine efficiency, e.g., via heap-based acceleration for repeated selection and sorting.

While A$_2$DEPT enables structural breakthroughs, open-ended synthesis also introduces system-level risks. We observed cases where the LLM introduced refactorings (e.g., set-to-bitmask replacements) that violated cross-module type consistency. Unlike template-bound AHD with fixed representations, open-ended AAD must preserve cross-module invariants under reconfiguration. Appendix F.2 analyzes these failures, where programs are syntactically complete but violate type contracts.

## 5. Discussion

A$_2$DEPT advances LLM-driven automated algorithm design through two complementary mechanisms. First, it reduces the burden of long-horizon reasoning by decomposing one-shot synthesis into a sequence of verifiable editing steps. This shifts the paradigm from generating entire algorithms to iteratively validating localized changes, significantly increasing the yield of executable candidates. Second, by separating stable components from mutable strategy code, A$_2$DEPT regularizes the search space. Constraining edits to the mutable region prevents accidental corruption of foundational utilities, allowing search effort to be concentrated on improving algorithmic logic.

Despite these advantages, A$_2$DEPT still has practical limitations. The closed-loop pipeline introduces overhead from repair and validation, motivating engineering optimizations such as batching, caching, and static checks. Moreover, A$_2$DEPT is currently designed for algorithmic-level solver synthesis, where candidate programs remain at the scale of modular research prototypes. Scaling open-ended synthesis to industrial solver codebases spanning many files and much larger code volumes would likely require additional architectural support, such as hierarchical module management, incremental compilation, and file-level dependency tracking. We view these extensions as promising directions for future work.

## 6. Conclusion

In this paper, we introduced A$_2$DEPT, an open-ended AAD framework that makes program-space evolution practical via localized program transformations. By coupling tree-based trajectory memory with feedback-driven operators and executability maintenance, A$_2$DEPT enables system-level solver redesign while keeping candidates runnable. These results suggest that shifting the search target from heuristic slots to complete executable programs is a viable and scalable paradigm for LLM-driven optimization. Across multiple combinatorial optimization benchmarks, A$_2$DEPT consistently outperforms representative LLM-based AHD baselines. Future work will focus on reducing verification cost, improving robustness to model reliability, and extending A$_2$DEPT to multi-objective settings.

## Impact Statement

This paper presents work whose goal is to advance the field of machine-learning research on evaluation-driven program search for optimization. We do not anticipate direct negative societal impacts from this work in its current research setting. As the method generates executable code, we recommend standard safeguards (e.g., sandboxed execution and testing) before any real-world deployment.

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

# A. Extended Background and Problem Formulation

In this section, we provide detailed explanations of four representative LLM-based AHD frameworks: FunSearch, EoH, ReEvo, and MCTS-AHD. These methods represent the state-of-the-art in leveraging LLMs as crossover and mutation operators within evolutionary search frameworks.

## A.1. Review of LLM-based AHD Frameworks

In the rapidly evolving landscape of AHD, LLMs have emerged as intelligent operators, replacing traditional stochastic mutation rules. This section provides an in-depth technical review of four representative frameworks that serve as the foundation and primary baselines for our work.

**1. FunSearch (Romera-Paredes et al., 2024) (Searching in the Function Space).** FunSearch pioneered the application of LLMs to open mathematical discovery by employing an *Island-Based Evolutionary Model*. The workflow proceeds cyclically through three distinct components: (1) A sampler selects high-scoring programs from distributed databases using a probabilistic method that favors correctness and conciseness; (2) An LLM mutator uses these programs as few-shot examples to generate new function bodies via semantic crossover; and (3) An evaluator executes the generated code, retaining only those that pass verification and improve upon the best known scores. Its core contribution lies in "Best-Shot Prompting," demonstrating that feeding an LLM its own best historical outputs creates a positive feedback loop for knowledge discovery.

**2. EoH (Liu et al., 2024a) (Evolution of Heuristics).** Addressing the "black-box" limitations of direct code generation, EoH introduces a *Dual-Evolution Paradigm* that decouples algorithmic reasoning from implementation. Instead of evolving code alone, it maintains a population of thought-code pairs, where Thoughts describe the strategy in natural language and Codes provide the implementation. The framework mimics Genetic Algorithms (GA) through three prompting operators: generational prompts for diverse initial ideas; mutation modifies existing logic; and crossover synthesizes advantages from two parent strategies. By enforcing a Think-then-Code process, EoH significantly enhances the semantic validity and interpretability of the generated heuristics.

**3. ReEvo (Ye et al., 2024) (Reflective Evolution).** ReEvo transforms the LLM from a passive generator into an active learner via a novel *reflexion mechanism*. Unlike traditional methods that discard failed individuals, ReEvo captures execution logs and error tracebacks to generate verbal gradients—textual feedback explaining *why* a heuristic failed. The process distinguishes between two types of learning: Short-term Reflection debugs syntax or runtime errors within the current iteration, while Long-term Reflection summarizes successful patterns into a persistent experience database. This mechanism allows the system to learn from mistakes, drastically improving sample efficiency compared to random search.

**4. MCTS-AHD (Zheng et al., 2025) (Monte Carlo Tree Search for AHD).** Identifying that population-based methods (like FunSearch and EoH) are prone to local optima, MCTS-AHD reformulates the heuristic design process as a global decision tree. It replaces linear evolution with a structured MCTS cycle: (1) Selection uses the Upper Confidence Bound (UCB) to balance exploration and exploitation; (2) Expansion prompts the LLM to generate child nodes; (3) Simulation evaluates the new heuristic; and (4) Backpropagation updates the statistics of all ancestor nodes. This structural shift enables the framework to revisit and develop temporarily underperforming branches that hold long-term potential, ensuring a more comprehensive exploration of the heuristic space.

## A.2. The Step-by-Step Construction

To validate the performance of AHD in addressing combinatorial optimization problems using LLMs, we adopt the Step-by-Step Construction paradigm as the general framework for AHD . Unlike specialized meta-heuristics that require designing complex, domain-specific operators, constructive methods rely on a minimal set of expert knowledge.

### A.2.1. FRAMEWORK CONCEPT

Step-by-step construction serves as an intuitive and universal framework capable of addressing a vast array of CO problems. Fundamentally, it considers the process of gradually extending a partial solution $s$ from scratch until a complete and feasible solution is constructed. In each step of the construction, the framework acts as a priority assigner: it evaluates all valid candidates (decision variables) based on the current state and assigns a numerical priority to each. The candidate with the

highest priority is then deterministically added to the solution. This cycle repeats until the solution satisfies all problem constraints and completeness requirements.

### A.2.2. GENERAL MATHEMATICAL FORMULATION

Formally, we model the solution process as a discrete trajectory $s_0 \rightarrow s_1 \rightarrow \cdots \rightarrow s_T$. For any CO problem instance $\mathcal{I}$, the construction is defined by a tuple $\langle \mathcal{S}, \Omega, \pi \rangle$:

- **State** ($s_t \in \mathcal{S}$): Represents the partial solution constructed up to step $t$. The initial state $s_0 = \emptyset$.

- **Candidate Set** ($\Omega(s_t)$): The set of all feasible decision variables available at state $s_t$ that satisfy problem constraints (e.g., unvisited cities in TSP, remaining items in Knapsack).

- **Priority Function** ($\pi$): This is the core heuristic optimized by AHD. It assigns a scalar score to each candidate $c \in \Omega(s_t)$ based on the current context:
$$\text{Score}(c) = \pi(c, s_t, \mathcal{I}; \theta), \tag{8}$$
where $\theta$ represents the logic or formula generated by the LLM.

The transition rule is deterministic and greedy with respect to the priority function:
$$s_{t+1} \leftarrow s_t \cup \{\arg \max_{c \in \Omega(s_t)} \text{Score}(c)\}, \tag{9}$$

This cycle repeats until $s_t$ is complete.

**Instantiation Example: CFLP.**  To illustrate how this abstract model translates into executable code, we consider CFLP.

The following pseudo-code demonstrates this mapping. The framework provides the outer loop, while the LLM generates the logic for `evaluate_candidate`.

---

**[Step-by-Step Skeleton for CFLP]**

```python
def solve_cflp(data):
    # 1. Initialization (s_0)
    current_solution = initialize()

    # 2. Construction Loop
    while not is_complete(current_solution):
        # Identify Candidates (\Omega)
        candidates = get_feasible_facilities(data, current_solution)

        best_cand = None
        best_score = -float('inf')

        # Evaluate Candidates using Priority Function (\pi)
        for cand in candidates:
            # =====================================================
            # THIS FUNCTION IS GENERATED BY THE LLM
            # Score(c) = \pi(c, s_t, I)
            # =====================================================
            score = evaluate_candidate(cand, current_solution, data)

            if score > best_score:
                best_score = score
                best_cand = cand

        # Transition (s_{t+1})
        if best_cand:
            current_solution.add(best_cand)
            update_capacities(current_solution)
```

---

```
        else:
            break

    return current_solution
```

### A.2.3. UNIFIED INTERFACE FOR AHD

To enable the LLMs to evolve a policy capable of approximating the optimal action-value function $Q^*(s_t, a)$, the function interface must provide a sufficient statistic of the decision process. If the heuristic lacks access to dynamic state variables, it becomes mathematically impossible to estimate future rewards.

Therefore, we define a standardized heuristic interface that exposes the necessary decision context. Regardless of the specific problem domain, the priority function $\pi$ generated by the LLM adheres to the following generic signature:

**[General Heuristic Template]**

```
def evaluate_candidate(
    candidate: Action,
    current_state: State,
    problem_instance: ProblemContext
) -> float:
    """
    The generic interface for the evolved heuristic priority function.
    Goal: Output a scalar score approximating Q*(s, a) to guide the greedy choice.

    Args:
        candidate: The specific action/move being evaluated (corresponds to c).
        current_state: Dynamic snapshot of the partial solution (corresponds to s_t).
        problem_instance: Static global data (corresponds to I).
    """
    # LLM implements logic here using mathematical operators or logical rules
    pass
```

### A.3. The Knowledge-Guided Local Search

To evaluate the generalization capability of AHD beyond constructive heuristics, we employ the Guided Local Search (GLS) paradigm. Unlike constructive methods that build solutions from scratch, GLS refines complete incumbent solutions through iterative local improvements, while injecting external guidance to avoid stagnation in poor local optima.

### A.3.1. FRAMEWORK CONCEPT

GLS serves as a hybrid paradigm that bridges learned heuristics with classical local search solvers. In our setting, GLS is instantiated as Knowledge-Guided Local Search (KGLS), where the guidance is explicitly represented as knowledge-based matrices generated by an LLM. Crucially, KGLS separates high-level guidance synthesis from low-level move execution: the LLM acts as a knowledge generator that produces state-conditioned guidance structures, while a fixed solver performs the actual neighborhood search using these structures to bias move selection, penalize undesirable patterns, or reshape the effective search landscape. This design allows the guidance to adapt as the incumbent solution evolves, enabling targeted escape from local optima without rewriting the underlying solver.

### A.3.2. GENERAL MATHEMATICAL FORMULATION

We consider minimizing an objective $f(x)$ over a feasible space $\mathcal{X}$ for a given instance $\mathcal{I}$. The KGLS process is defined by the interaction between a knowledge generator $\Psi$ and a guided solver $\Phi$:

- **Problem Instance ($\mathcal{I}$):** Static data describing the optimization task.

- **Incumbent State ($x_t$):** A complete feasible solution maintained by the local search at iteration $t$.

- **Knowledge Structure ($K_t$):** A structured guidance signal encoding the desirability of solution components or moves under the current search context.

- **Knowledge Generator ($\Psi$):** The core logic optimized by AHD. It produces state-conditioned guidance:

$$K_t = \Psi(\mathcal{I}, x_t; \theta), \tag{10}$$

  where $\theta$ denotes the program (code) synthesized by the LLM.

- **Guided Local Search Solver ($\Phi$):** A fixed local search routine that uses $K_t$ to bias its neighborhood exploration:

$$x_{t+1} \leftarrow \Phi(\mathcal{I}, x_t, K_t). \tag{11}$$

Unlike purely offline guidance that is computed once before search, KGLS updates the guidance online: $\Psi$ can be queried periodically to adapt $K_t$ to the evolving incumbent solution. The LLM therefore influences *how* the solver searches, rather than directly proposing full solutions.

**Instantiation Example.** To illustrate this paradigm, we consider CVRP. The LLM generates code for a knowledge generator that outputs an edge-priority matrix conditioned on the current route configuration. The solver then uses this matrix to bias perturbations or local moves, prioritizing promising edges or discouraging repeatedly harmful structures.

---

**[KGLS framework for CVRP]**

```python
def solve_cvrp_kgls(data):
    x = random_initialization(data)

    for t in range(T):
        # =========================================
        # THIS FUNCTION IS GENERATED BY THE LLM
        # K_t = Psi(I, x_t)
        # =========================================
        K = get_knowledge_matrix(data, x)

        # Fixed guided local search step:
        # use K to bias move selection / penalties
        x = guided_local_search_step(data, x, guidance=K)

    return x
```

---

### A.3.3. UNIFIED INTERFACE FOR KGLS

To enable the LLM to capture *both* global structure and the current search context, the interface must expose (i) the static instance data and (ii) a sufficient summary of the incumbent solution / solver state. In contrast to constructive interfaces that operate on partial solutions, the KGLS interface operates on complete incumbents and outputs structured guidance for the next local-search step.

We define a standardized interface for the knowledge generator. Across problem domains, the function $\Psi$ synthesized by the LLM follows the generic signature:

---

**[General Knowledge Generator Template]**

```python
def get_knowledge_matrix(
    problem_instance: ProblemContext,
    incumbent_solution: SolutionState
) -> np.ndarray:
    """
    Generic interface for state-conditioned knowledge generation in KGLS.
    Goal: Output a matrix/tensor that biases local search moves given (I, x_t).
```

---

```
    Args:
        problem_instance: Static global data (coordinates, demands, costs, etc.).
        incumbent_solution: Current complete solution (or a compact state summary).

    Returns:
        A NumPy array (e.g., N x N matrix or a higher-order tensor) encoding
        scores/priorities/penalties used by the fixed local-search solver.
    """
    pass
```

### A.3.4. FIXED LOCAL SEARCH OPERATORS IN KGLS

In KGLS, the LLM only synthesizes a guidance structure $K_t = \Psi(\mathcal{I}, x_t)$, while the search dynamics are governed by a fixed local-search routine $\Phi$. Each iteration follows a unified loop: (i) query $\Psi$ to obtain $K_t$, (ii) apply a guidance-driven perturbation to escape the current basin, and (iii) run a short bounded first-improvement local search to reach the next incumbent.

**CFLP.** The incumbent is an assignment vector $a \in \{1, \ldots, n_{\text{fac}}\}^{n_{\text{cust}}}$ (with $-1$ for unassigned). $\Phi$ uses a relocate neighborhood that moves a customer $c$ from $f_{\text{old}}$ to $f_{\text{new}}$ if capacity allows, evaluated by a marginal-cost delta including transport and facility opening/closing costs. Perturbation uses $K_t \in \mathbb{R}^{n_{\text{fac}} \times n_{\text{cust}}}$ to force a few high-scoring (facility, customer) reassignments.

**CVRP.** The incumbent is a set of routes. $\Phi$ applies bounded first-improvement with two neighborhoods: intra-route 2-opt and inter-route relocate under capacity constraints. $K_t \in \mathbb{R}^{N \times N}$ is an edge-priority matrix; perturbation selects high-scoring absent edges $(u, v)$ and enforces them by relocating $v$ to follow $u$ when feasible.

**FJSP.** We search in the machine-assignment subspace, where the incumbent maps each operation to a compatible machine. Schedules are induced by an earliest-start simulator. $\Phi$ performs stochastic reassignment moves on sampled operations, accepting only makespan-improving changes. $K_t \in \mathbb{R}^{n_{\text{ops}} \times n_{\text{mach}}}$ scores operation–machine pairs; perturbation forces a few top-scoring reassignment decisions while masking incompatible machines.

**MIS.** The incumbent is a boolean mask over vertices. $\Phi$ greedily adds feasible vertices to obtain a maximal independent set. The guidance is a score vector $K_t \in \mathbb{R}^{|V|}$; perturbation inserts several high-scoring unselected vertices and repairs feasibility by removing conflicting neighbors.

These lightweight, problem-specific neighborhoods provide a stable backbone; KGLS improves performance by evolving $\Psi$, i.e., by making $K_t$ induce more effective perturbations and move biases.

### A.4. Formal Definitions of Optimization Problems

In this section, we provide the mathematical formulations for the six NP-hard problems investigated in this paper.

### A.4.1. CAPACITATED FACILITY LOCATION PROBLEM

In this study, we address the Capacitated Facility Location Problem (Kuehn & Hamburger, 1963). Unlike the general split-delivery version, CFLP requires that each customer's demand be satisfied by exactly one facility, introducing strict combinatorial constraints that make the problem strongly NP-hard.

**Notations and Variables.** Let $I = \{1, \ldots, m\}$ denote the set of potential facility locations and $J = \{1, \ldots, n\}$ be the set of customers. The problem parameters are: $f_i$ as the fixed setup cost for opening facility $i$; $q_i$ as the capacity of facility $i$; $d_j$ as the demand of customer $j$; and $c_{ij}$ as the transportation cost serving customer $j$ from facility $i$. We define two sets of binary decision variables:

- $y_i \in \{0, 1\}$: Equals 1 if facility $i$ is opened, 0 otherwise.

- $x_{ij} \in \{0, 1\}$: Equals 1 if customer $j$ is assigned to facility $i$, 0 otherwise.

**Mathematical Formulation.**  The objective is to minimize the total cost, comprising fixed opening costs and variable transportation costs. The Integer Linear Programming (ILP) model (Avella & Boccia, 2009a) is formulated as:

$$\text{Minimize} \quad Z = \sum_{i \in I} f_i y_i + \sum_{i \in I} \sum_{j \in J} c_{ij} x_{ij}, \tag{12}$$

$$\text{Subject to} \quad \sum_{i \in I} x_{ij} = 1, \quad \forall j \in J, \tag{13}$$

$$\sum_{j \in J} d_j x_{ij} \leq q_i y_i, \quad \forall i \in I, \tag{14}$$

$$x_{ij}, y_i \in \{0, 1\}, \quad \forall i \in I, j \in J. \tag{15}$$

**Constraint Analysis.**  Constraint (13) enforces the single-source property, ensuring every customer is served by exactly one facility. Constraint (14) functions as a critical linking constraint. It simultaneously enforces capacity limits ($\sum d_j \leq q_i$) and logical consistency: if a facility is closed ($y_i = 0$), the right-hand side becomes 0, forcing all $x_{ij}$ to 0. This coupling of location and allocation decisions creates a complex search landscape combining the hardness of the Knapsack Problem and Set Covering Problem.

A.4.2. CAPACITATED VEHICLE ROUTING PROBLEM

The CVRP (Dantzig & Ramser, 1959) is a generalization of TSP. We consider the standard version involving a homogeneous fleet of vehicles based at a single depot, where split deliveries are not permitted.

Let $G = (V, A)$ be a complete directed graph, where $V = \{0\} \cup C$ denotes the vertex set comprising a central depot 0 and a set of customers $C = \{1, \ldots, n\}$. Each customer $i \in C$ has a non-negative demand $d_i$, and each arc $(i, j) \in A$ is associated with a travel cost $c_{ij}$. The fleet consists of a set $K$ of identical vehicles, each constrained by a maximum capacity $Q$. To model the vehicle trajectories, we define the binary decision variable $x_{ij}^k$, which takes the value 1 if vehicle $k \in K$ travels directly from node $i$ to node $j$, and 0 otherwise.

**Mathematical Formulation.**  The objective is to minimize the total fleet travel distance while satisfying demand and operational constraints. The formulation (Lysgaard et al., 2004) is given as:

$$\text{Minimize} \quad Z = \sum_{k \in K} \sum_{(i,j) \in A} c_{ij} x_{ij}^k, \tag{16}$$

$$\text{Subject to} \quad \sum_{k \in K} \sum_{j \in V} x_{ij}^k = 1, \quad \forall i \in C, \tag{17}$$

$$\sum_{i \in V} x_{ih}^k - \sum_{j \in V} x_{hj}^k = 0, \quad \forall h \in C, \forall k \in K, \tag{18}$$

$$\sum_{i \in C} d_i \sum_{j \in V} x_{ij}^k \leq Q, \quad \forall k \in K, \tag{19}$$

$$x_{ij}^k \in \{0, 1\}, \quad \forall (i, j) \in A, k \in K. \tag{20}$$

The constraints enforce the structural validity of the routes through three distinct mechanisms. First, Constraint (17) mandates that every customer is visited exactly once by exactly one vehicle, strictly prohibiting split deliveries. Second, Constraint (18) ensures flow conservation and route continuity; it dictates that if vehicle $k$ enters node $h$, it must also depart from node $h$, which, combined with the depot definition, necessitates the formation of closed tours starting and ending at node 0. Finally, Constraint (19) limits the total load on any route to the capacity $Q$.

### A.4.3. FLEXIBLE JOB SHOP SCHEDULING PROBLEM

The Flexible Job Shop Scheduling Problem (Brucker & Schlie, 1990)is a generalization of the classical Job Shop Scheduling Problem. It introduces an additional layer of complexity by allowing each operation to be processed on any machine from a given compatible set. Consequently, solving FJSP requires addressing two coupled sub-problems: Machine Assignment (Routing) and Operation Sequencing (Scheduling).

Let $J = \{1, \dots, n\}$ denote the set of independent jobs and $M = \{1, \dots, m\}$ denote the set of available machines. Each job $i \in J$ requires the completion of a sequential chain of operations $O_{i,1}, O_{i,2}, \dots, O_{i,n_i}$. For any specific operation $O_{ij}$, let $M_{ij} \subseteq M$ represent the subset of compatible machines capable of processing it, where the processing time on a specific machine $k \in M_{ij}$ is given by $p_{ijk}$. The problem formulation relies on two types of decision variables to capture both assignment and scheduling decisions: the binary variable $x_{ijk} \in \{0, 1\}$ takes the value 1 if operation $O_{ij}$ is assigned to machine $k$ (and 0 otherwise), while the continuous variable $S_{ij} \geq 0$ specifies the start time of operation $O_{ij}$.

**Mathematical Formulation.** The objective is to minimize the makespan ($C_{\max}$), defined as the completion time of the last operation of the last job. The Mixed-Integer Linear Programming (MILP) formulation (Chen et al., 1999) is:

$$\text{Minimize} \quad C_{\max}, \tag{21}$$

$$\text{Subject to} \quad C_{\max} \geq S_{i,n_i} + \sum_{k \in M_{i,n_i}} p_{i,n_i,k} x_{i,n_i,k}, \quad \forall i \in J \tag{22}$$

$$\sum_{k \in M_{ij}} x_{ijk} = 1, \quad \forall i \in J, \forall j \tag{23}$$

$$S_{i,j+1} \geq S_{ij} + \sum_{k \in M_{ij}} p_{ijk} x_{ijk}, \quad \forall i \in J, \forall j < n_i \tag{24}$$

$$S_{ij} + p_{ijk} \leq S_{uv} \lor S_{uv} + p_{uvk} \leq S_{ij}, \quad \forall (i,j) \neq (u,v) \text{ on } k. \tag{25}$$

The constraints govern the physical and logical flow of the workshop. Constraint (23) mandates valid machine assignment, ensuring each operation is processed by exactly one machine selected from its compatible set $M_{ij}$. Regarding the technological sequence, Constraint (24) enforces job precedence, dictating that an operation $O_{i,j+1}$ cannot commence until its predecessor $O_{ij}$ has finished. Finally, Constraint (25) imposes disjunctive capacity limits (simplified here for brevity), ensuring that a machine processes at most one operation at a time; specifically, if two operations $O_{ij}$ and $O_{uv}$ are assigned to the same machine $k$, their execution intervals must be non-overlapping.

### A.4.4. MAXIMUM INDEPENDENT SET

The Maximum Independent Set problem (Karp, 2009) is a fundamental challenge in graph theory and combinatorial optimization. Unlike the routing or scheduling problems defined above, MIS represents a pure topological optimization task, focusing on resolving structural conflicts within a network.

Let $G = (V, E)$ be an undirected graph, where $V = \{1, \dots, n\}$ denotes the set of vertices and $E \subseteq V \times V$ represents the set of edges capturing conflicts or connections between vertices. To determine the composition of the independent set, we define a binary decision variable $x_i \in \{0, 1\}$ for each vertex $i \in V$, which equals 1 if vertex $i$ is selected for the independent set, and 0 otherwise.

**Mathematical Formulation.** The objective is to maximize the cardinality of the selected set subject to the independence property. The formulation (Wolsey & Nemhauser, 1999) is:

$$\text{Maximize} \quad Z = \sum_{i \in V} x_i, \tag{26}$$

$$\text{Subject to} \quad x_i + x_j \leq 1, \quad \forall (i,j) \in E, \tag{27}$$

$$x_i \in \{0, 1\}, \quad \forall i \in V. \tag{28}$$

Constraint (27) is the core independence constraint (also known as the edge conflict constraint). It enforces that for any pair of adjacent vertices $(i, j)$ connected by an edge, at most one of them can be included in the solution set. This creates a rigorous selection pressure where choosing a high-degree vertex may invalidate a large number of its neighbors, requiring the algorithm to strategically prioritize nodes that contribute to the objective without causing excessive "collateral damage" to the feasible space.

### A.4.5. CAPACITATED ELECTRIC VEHICLE ROUTING PROBLEM WITH TIME WINDOWS

The CEVRPTW(Schneider et al., 2014) extends the classic VRPTW by incorporating limited battery capacity and the possibility of recharging at designated stations. It represents a multi-attribute constrained problem requiring the simultaneous management of capacity, battery energy, and time windows.

The problem is defined on a directed graph $G = (V, A)$, where the vertex set $V = \{0, N+1\} \cup C \cup F'$ comprises the depot (represented by start node 0 and end node $N + 1$), a set of customers $C = \{1, \ldots, N\}$, and a set of dummy nodes $F'$ representing recharging stations to accommodate multiple visits. Key problem parameters include the vehicle's maximum cargo capacity $Q$ and battery capacity $B$. Each node $i$ is associated with a demand $d_i$, a service or recharging duration $s_i$, and a time window $[e_i, l_i]$. Furthermore, traversing an arc $(i, j) \in A$ incurs a travel cost $c_{ij}$ and consumes energy $h_{ij}$. To model the routing plan, we employ the binary decision variable $x_{ij} \in \{0, 1\}$, which equals 1 if arc $(i, j)$ is traversed. Additionally, continuous variables are defined to track the system state: $\tau_i$ denotes the arrival time, $u_i$ represents the remaining cargo load, and $y_i$ indicates the remaining battery level upon arrival at node $i$.

**Mathematical Formulation.**   The objective minimizes the total travel cost. Let $K$ denote the number of vehicles, and let $s_i$ denote the service time at customers (and the effective charging time at stations). We adopt the full-recharge policy(Schneider et al., 2014), assuming vehicles always recharge to full capacity $B$ upon visiting a station. The formulation is:

$$\text{Minimize} \quad \sum_{(i,j) \in A} c_{ij}\, x_{ij}, \tag{29}$$

$$\text{Subject to} \quad \sum_{j:(i,j) \in A} x_{ij} = 1, \quad \forall i \in C, \tag{30}$$

$$\sum_{i:(i,j) \in A} x_{ij} = 1, \quad \forall j \in C, \tag{31}$$

$$\sum_{j:(0,j) \in A} x_{0j} = K, \qquad \sum_{i:(i,N+1) \in A} x_{i,N+1} = K, \tag{32}$$

$$\sum_{j:(i,j) \in A} x_{ij} - \sum_{j:(j,i) \in A} x_{ji} = 0, \quad \forall i \in V \setminus \{0, N+1\}, \tag{33}$$

$$\sum_{j:(i,j) \in A} x_{ij} \leq 1, \quad \sum_{j:(j,i) \in A} x_{ji} \leq 1, \quad \forall i \in F', \tag{34}$$

$$u_j \leq u_i - d_i x_{ij} + Q(1 - x_{ij}), \quad \forall (i,j) \in A, \tag{35}$$

$$\tau_j \geq \tau_i + s_i + t_{ij} - M(1 - x_{ij}), \quad \forall (i,j) \in A, \tag{36}$$

$$e_i \leq \tau_i \leq l_i, \quad \forall i \in V, \tag{37}$$

$$y_j \leq y_i - h_{ij} x_{ij} + B(1 - x_{ij}), \quad \forall i \in C \cup \{0\}, \forall (i,j) \in A, \tag{38}$$

$$y_j \leq B - h_{ij} x_{ij} + B(1 - x_{ij}), \quad \forall i \in F', \forall (i,j) \in A, \tag{39}$$

$$x_{ij} \in \{0, 1\}, \; \tau_i \geq 0, \; u_i \geq 0, \; y_i \geq 0. \tag{40}$$

Constraints (35) enforce capacity feasibility. Constraints (36)–(37) impose time-window feasibility. Constraints (38) and (39) govern battery dynamics under the full-recharge assumption: for arcs leaving a customer or depot ($i \in C \cup \{0\}$), the remaining energy at $j$ is derived from the arrival energy at $i$; conversely, for arcs leaving a charging station ($i \in F'$), the vehicle departs with full capacity $B$, regardless of its arrival level.

A.4.6. MULTI-MODE RESOURCE-CONSTRAINED PROJECT SCHEDULING PROBLEM

The MRCPSP (Talbot, 1982) represents the most generalized and challenging form of project scheduling. It extends the standard RCPSP by allowing each activity to be executed in one of multiple modes. Each mode represents a distinct trade-off between processing time and resource consumption (e.g., "fast but expensive" vs. "slow but cheap"), requiring simultaneous optimization of activity mode selection and scheduling sequences.

We define a project as a directed acyclic graph $G = (V, E)$, where $V = \{1, \ldots, J\}$ denotes the set of activities. Activities 1 and $J$ are dummy start and end nodes with zero duration and resource consumption. The project utilizes a set $\mathcal{R}$ of renewable resources and a set $\mathcal{N}$ of non-renewable resources. Each resource $r \in \mathcal{R} \cup \mathcal{N}$ has a capacity $K_r$. Each activity $j$ supports a set of execution modes $M_j$; selecting a mode $m \in M_j$ determines the activity's duration $d_{jm}$ as well as its consumption $k_{jmr}$ of resource $r$.

**Mathematical Formulation.** The problem is formulated using the discrete time-indexed binary decision variable $x_{jmt}$, which equals 1 if activity $j$ is performed in mode $m$ and finishes at the end of period $t$, and 0 otherwise. Let $EFT_j$ and $LFT_j$ denote the earliest and latest finish times for activity $j$, derived from critical path analysis. The objective is to minimize the project makespan, determined by the completion time of the dummy sink activity $J$. The formulation (Kolisch & Drexl, 1997) is:

$$\text{Minimize} \quad \sum_{t=EFT_J}^{LFT_J} t \cdot x_{J1t}, \tag{41}$$

$$\text{Subject to} \quad \sum_{m=1}^{M_j} \sum_{t=EFT_j}^{LFT_j} x_{jmt} = 1, \quad \forall j = 1, \ldots, J, \tag{42}$$

$$\sum_{m=1}^{M_i} \sum_{t=EFT_i}^{LFT_i} t \cdot x_{imt} \leq \sum_{m=1}^{M_j} \sum_{t=EFT_j}^{LFT_j} (t - d_{jm}) x_{jmt}, \quad \forall (i,j) \in E, \tag{43}$$

$$\sum_{j=2}^{J-1} \sum_{m=1}^{M_j} k_{jmr} \sum_{\tau=t}^{t+d_{jm}-1} x_{jm\tau} \leq K_r, \quad \forall r \in \mathcal{R}, \forall t \in \mathcal{T}, \tag{44}$$

$$\sum_{j=2}^{J-1} \sum_{m=1}^{M_j} k_{jmr} \sum_{t=EFT_j}^{LFT_j} x_{jmt} \leq K_r, \quad \forall r \in \mathcal{N}, \tag{45}$$

$$x_{jmt} \in \{0, 1\}. \tag{46}$$

Constraint (42) ensures that every activity $j$ is assigned exactly one mode and a unique valid finish time within its time window $[EFT_j, LFT_j]$. Constraint (43) enforces the precedence relations: the finish time of a predecessor $i$ (LHS) must not exceed the start time of its successor $j$ (RHS, calculated as finish time $t$ minus duration $d_{jm}$). The resource constraints differentiate between renewable resources ($\mathcal{R}$) and non-renewable resources ($\mathcal{N}$). Constraint (44) guarantees that for each renewable resource $r$ and time period $t$, the total consumption by all active activities does not exceed the per-period capacity $K_r$. The inner summation $\sum_{\tau=t}^{t+d_{jm}-1} x_{jm\tau}$ correctly identifies activities that are in progress at time $t$ by considering all possible finish times $\tau$ that would imply the activity covers period $t$. Finally, Constraint (45) restricts the global consumption of non-renewable resources (e.g., budget) to the total available limit $K_r$.

# B. Implementation Details of A$_2$DEPT

This section provides the technical details required to reproduce the A$_2$DEPT framework, presenting the complete algorithmic workflow and key execution logic.

Algorithm 1 provides the pseudo-code of A$_2$DEPT, including the overall workflow and key hyperparameter settings. The procedure consists of an initialization stage (Line 1–4) followed by an iterative tree-expansion loop (Line 5–41) until the LLM-call budget $T_{\max}$ is exhausted. In each iteration, A$_2$DEPT (i) constructs the parent set via hybrid selection—Simulated Annealing on the frontier (Line 6–7) and Boltzmann supplementation from the historical tree when needed (Line 8–11), (ii)

selects operators/parents and generates a candidate program with one LLM call (Line 12–17), (iii) enforces executability through a closed-loop dependency repair process that may trigger additional LLM calls (Line 18–22), and (iv) prunes/builds and evaluates the program, then updates the search tree, operator weights, and annealing schedule (Line 23–40). The algorithm returns the best program $P^*$ found (Line 42).

---

**Algorithm 1** A$_2$DEPT: Adaptive Algorithm Design via Evolutionary Program Trees

---

**Input:** Evaluation dataset $D$, init size $K$, LLM-call budget $T_{\max}$, operators $\mathcal{O}=\{m_1, m_2, e_1\}$, temperature $T_0$, decay $\alpha$, re-anneal $\Delta T$, stall threshold $N_{\text{stall}}$.
**Output:** Best program $P^*$.
1: $t \leftarrow 0$; $T_{\text{curr}} \leftarrow T_0$; $\Delta t_{\text{stall}} \leftarrow 0$ {$t$: number of LLM calls}
2: $\mathcal{T} \leftarrow$ COT-COLDSTART($K$) {init roots; App. C.1}
3: $t \leftarrow t +$ CALLS(COLDSTART)
4: $P^* \leftarrow \arg\max_{n \in \mathcal{T}} S(n)$
5: **while** $t < T_{\max}$ **do**
6: $\quad \mathcal{F} \leftarrow$ GETFRONTIER($\mathcal{T}$)
7: $\quad \mathcal{P}_{next} \leftarrow$ SASELECT($\mathcal{F}, T_{\text{curr}}$) {Metropolis vs. parent}
8: $\quad$ **if** $|\mathcal{P}_{next}| < K$ **then**
9: $\quad\quad \mathcal{H} \leftarrow \mathcal{T} \setminus \mathcal{F}$ {history only}
10: $\quad\quad \mathcal{P}_{next} \leftarrow \mathcal{P}_{next} \cup$ BOLTZMANNSAMPLE($\mathcal{H}, K-|\mathcal{P}_{next}|, T_{\text{curr}}$)
11: $\quad$ **end if**
12: $\quad$ **for each** $n \in \mathcal{P}_{next}$ **do**
13: $\quad\quad$ **if** $t \geq T_{\max}$ **then**
14: $\quad\quad\quad$ **break**
15: $\quad\quad$ **end if**
16: $\quad\quad (op, \mathcal{X}) \leftarrow$ SELECTOPPARENTS($n, \mathcal{T}$) {$e_1$ is triggered conditionally; else sample $m_1/m_2$ by $\boldsymbol{\omega}_n$}
17: $\quad\quad Code \leftarrow$ LLM-GEN($op, \mathcal{X}$); $t \leftarrow t + 1$
18: $\quad\quad \mathcal{U} \leftarrow$ GETMISSINGDEPS($Code$)
19: $\quad\quad$ **while** $\mathcal{U} \neq \emptyset$ **and** $t < T_{\max}$ **do**
20: $\quad\quad\quad Code \leftarrow Code \cup$ LLM-IMPL($\mathcal{U}$); $t \leftarrow t + 1$
21: $\quad\quad\quad \mathcal{U} \leftarrow$ GETMISSINGDEPS($Code$)
22: $\quad\quad$ **end while**
23: $\quad\quad P_{\text{new}} \leftarrow$ BUILD(PRUNEDEADCODE($Code$))
24: $\quad\quad S_{\text{new}} \leftarrow$ EVALUATE($P_{\text{new}}, D$)
25: $\quad\quad$ **if** EVALFAILED **or** INFEASIBLE **then**
26: $\quad\quad\quad S_{\text{new}} \leftarrow -\infty$
27: $\quad\quad$ **end if**
28: $\quad\quad \mathcal{T} \leftarrow \mathcal{T} \cup \{(P_{\text{new}}, S_{\text{new}}, \text{PARENT}=n, \text{OP}=op)\}$
29: $\quad\quad \boldsymbol{\omega}_n \leftarrow$ UPDATEWEIGHTS($\boldsymbol{\omega}_n, op, S_{\text{new}}, S(n)$)
30: $\quad\quad$ **if** $S_{\text{new}} > S(P^*)$ **then**
31: $\quad\quad\quad P^* \leftarrow P_{\text{new}}$; $\Delta t_{\text{stall}} \leftarrow 0$
32: $\quad\quad$ **else**
33: $\quad\quad\quad \Delta t_{\text{stall}} \leftarrow \Delta t_{\text{stall}} + 1$
34: $\quad\quad$ **end if**
35: $\quad\quad$ **if** $\Delta t_{\text{stall}} \geq N_{\text{stall}}$ **then**
36: $\quad\quad\quad T_{\text{curr}} \leftarrow T_{\text{curr}} + \Delta T$; $\Delta t_{\text{stall}} \leftarrow 0$
37: $\quad\quad$ **else**
38: $\quad\quad\quad T_{\text{curr}} \leftarrow \alpha \cdot T_{\text{curr}}$
39: $\quad\quad$ **end if**
40: $\quad$ **end for**
41: **end while**
42: **return** $P^*$

---

# C. Prompt Engineering and Templates

This section documents the specific prompt templates employed across the A$_2$DEPT. To ensure transparency and reproducibility, we present the templates in the chronological order of the algorithm's execution.

In the templates below, placeholders like $\{\{$PROBLEM_DESC$\}\}$ indicate dynamic content injected at runtime.

## C.1. Phase I: Chain-of-Thought Cold Start

The initialization phase utilizes a three-stage pipeline to generate diverse and high-quality initial seeds.

### C.1.1. STEP 1: DEEP PROBLEM ANALYSIS

Before generating any strategies, we first prompt the LLM to perform a comprehensive domain analysis. This step ensures the subsequent design is grounded in mathematical rigor rather than surface-level pattern matching.

---

**[Template I-1: Deep Problem Analysis]**

You are an expert in Operations Research and algorithm design, acting as a senior consultant. Your task is to analyze the following optimization problem and provide a comprehensive implementation plan.

**Task Description:**
{{task_description}}
Problem Taxonomy:
- What is the fundamental nature of this problem? (e.g., Routing, Scheduling.). - Is it a standard problem or a variant?
Decision Variables:
- Define the decisions to be made mathematically. - Specify the domain.
Objective Function:
- What is the goal? (Minimize Cost, Maximize Profit.).
Constraints:
- Conditions that MUST be met for a solution to be valid. - Conditions that can be violated with a penalty.
Methodology Analysis: - Propose several potential algorithmic approaches to solve this.

Do NOT write any final Python code yet. Your entire output should be a structured analysis.

---

### C.1.2. STEP 2: SEQUENTIAL STRATEGY GENERATION

We generate the $k$-th strategy conditioned on the initial problem analysis and the summary of the previous $k-1$ generated plans. By explicitly inputting the history of existing strategies, we instruct the LLM to propose a new approach that is distinct from the previous ones, thereby ensuring the diversity of the initial population.

---

**[Template I-2: Sequential Strategy Generation]**

You are an expert in heuristic design and diversity optimization. Your goal is to generate a new solution strategy that is structurally different from the existing ones.

Problem Analysis:
{{ANALYSIS_RESULT}}
Previous $k-1$ Strategies:
{{HISTORY_SUMMARIES}}

**Task:**
Based on the analysis above, propose a NEW algorithmic strategy.
Output a short, descriptive identifier and a description of the algorithm steps.

Do NOT write any final Python code yet.

---

### C.1.3. STEP 3: SEED IMPLEMENTATION

This step translates the abstract text plan into the initial executable code skeleton, following the provided function interface.

---

**[Template I-3: Seed Implementation]**

You are a Senior Python Engineer specializing in algorithmic implementation. Your task is to translate a high-level algorithmic strategy into a concrete Python implementation, strictly adhering to the provided code skeleton.

**Task Description**: {{task_description}}

---

Algorithm Strategy:
{{STRATEGY_PLAN}}
Function Template:
{{FUNCTION_TEMPLATE}}

**Task:**
Implement the **Selected Strategy** using the structure defined in the Function Template.
Translate the strategy into Python code.
You MUST use the function signatures provided in the template.

Describe your new algorithm and main steps in one sentence. The description must be inside within boxed {{}}.
Do NOT give additional explanations.

## C.2. Phase II: Evolutionary Operators

In the evolutionary loop, A$_2$DEPT dynamically schedules operators based on feedback.

### C.2.1. OPERATOR 1: MICRO-TUNING ($m_1$)

The Micro-Tuning operator targets the *Mutable Strategies* ($\mathcal{F}_{mut}$) within the function registry. It focuses on refining the internal logic of a specific function—such as adjusting heuristic coefficients or enhancing tie-breaking mechanisms—to improve local efficiency without altering the global algorithmic flow or function signatures.

**[Template II-1: Micro-Tuning ($m_1$)]**

You are a Code Optimization Specialist. Your task is to refine the implementation of a specific algorithmic component to enhance its performance or solution quality.

**Task Description**: {{task_description}}
Current Implementation: {{FUNC_CODE}}

**Task:**
Optimizing the function shown above. Refine the internal logic to improve efficiency or effectiveness.
You can assume any helper functions called inside the original code are valid and available. Do NOT attempt to implement or modify them.
Return ONLY the optimized Python code for this specific function.

Do NOT change the function name, input arguments, or return type. The interface must remain exactly as is.
Do NOT give additional explanations.

### C.2.2. OPERATOR 2: MACRO-MUTATION ($m_2$)

This operator targets the *Algorithmic Backbone* (the main entry point). It encourages a top-down redesign by allowing the LLM to invoke abstract, undefined helper functions, which are automatically implemented by the system in subsequent steps.

**[Template II-2: Macro-Mutation ($m_2$)]**

You are an Expert Algorithm Architect. Your goal is to redesign the strategy to break out of local optima. You employ a **Top-Down Design** philosophy: focus on the flow, delegate details to sub-functions.

**Task Description**: {{task_description}}
Diagnosis / Previous Thought: {{PREVIOUS_THOUGHT}}
Current Function: {{CURRENT_CODE}}

**Task:**
Rewrite the Main Function to implement a totally new strategy from the given one.
You are free to call new, undefined helper functions.
Do NOT implement these new helpers yet. Just call them in your main logic.

Undefined functions will be implemented later.

Output ONLY the new Main Function code. Do NOT implement the helper functions you call.
If the Main Function requires standard libraries (e.g., 'import random', 'import math'), include them.
Describe your new algorithm and main steps in one sentence. The description must be inside within boxed {{}}.
Do NOT give additional explanations.

## C.2.3. OPERATOR 3: CROSSOVER ($e_1$)

This operator fuses two distinct parent algorithms based on their semantic roles. It extracts the structural backbone from one parent and the heuristic details from the other to synthesize a hybrid main strategy.

---

**[Template II-3: Semantic Crossover ($e_1$)]**

You are an Expert Integration Architect. Your goal is to synthesize a hybrid algorithm by merging the most effective logic/strategies from two parent algorithms. You employ a **Top-Down Design** philosophy: focus on the flow, delegate details to sub-functions.

**Task Description:**
{{task_description}}
Parent A:
Strategy: {{THOUGHT_A}}
Code: {{CODE_A_MAIN}}
Parent B:
Strategy: {{THOUGHT_B}}
Code: {{CODE_B_MAIN}}

**Task:** Create a **Hybrid Main Function** that combines the strengths of Parent A and Parent B.
You are free to call new, undefined helper functions.
Do NOT implement these new helpers yet. Just call them in your main logic.
Undefined functions will be implemented later.

Output ONLY the new Main Function code. Do NOT implement the helper functions you call.
If the Main Function requires standard libraries (e.g., 'import random', 'import math'), include them.
Describe your new algorithm and main steps in one sentence. The description must be inside within boxed {{}}.
Do NOT give additional explanations.

---

## C.2.4. ROLE-BASED FUNCTION PARTITIONING

This prompt implements the *function role analysis* step in A$_2$DEPT's Structured Program Representation. It is triggered only after backbone-level edits to the main function—i.e., when applying the top-down operators $m_2$ or $e_1$—to re-partition the function registry into mutable heuristic strategies and immutable problem definitions. Only the mutable subset is exposed to subsequent micro-tuning operators.

---

**[Template II-4: Function Role Analysis (Mutable Strategy Identification)]**

You are a lead algorithm architect. Your task is to analyze the function structure of the current algorithm and identify which functions are **mutable heuristic strategies** that can be optimized.

**Task Description**: {{task_description}}
Structure of the current algorithm (Signatures & Docstrings only): {{code_structure}}

**Task:** Classify functions into two roles:

- **Mutable Strategy**: heuristics, scoring rules, local search operators, selection/scheduling logic, move operators.

- **Immutable Definition**: feasibility/constraint checks, distance/cost calculations, data loading, basic utilities.

Return a JSON list containing **only** the **Mutable Strategy** functions.

**Output Format:** Return a JSON list of objects. Each object must include:

---

- `name`: function name

- `reason`: brief justification (5–10 words)

Only return the JSON list. Do not include additional explanations or markdown.

## C.2.5. AUTOMATED DEPENDENCY GOVERNANCE

This prompt serves as A$_2$DEPT's self-healing mechanism. It is triggered automatically whenever the generated or edited program contains missing dependencies (e.g., undefined helper functions) detected by static analysis or execution errors (such as `NameError`). Given the call context, the model is prompted to implement the required function(s) so that the program becomes executable.

---

**[Template II-5:   Dependency Repair (In-filling)]**

You are a Python Implementation Specialist. Your system has detected a `NameError`. Your task is to implement the missing dependency to make the code executable.

**Task Description**: {{task_description}}
Context (Code calling the missing function): {{MAIN_FUNC_CODE}}

**Task:** The code context above calls the helper function {{missing_func_name}}, but its definition is missing.
Please implement this specific function.

Analyze the "Context" code to deduce the required arguments and return type of the missing function.
If the Main Function requires standard libraries (e.g., 'import random', 'import math'), include them.
ONLY return the code for {{missing_func_name}}.
Do not give additional explanations.

---

## C.3. Problem Specifications and Interface Templates

To demonstrate the versatility of our framework, we apply it to two distinct combinatorial optimization domains: CFLP and CEVRPTW. Below, we present the function templates used for implementation.

## C.3.1. CFLP

---

**[Code Template: {{function_template}}]**

```python
def solve_cflp(
    facility_capacities: List[int],
    customer_demands: List[int],
    assignment_costs: List[List[int]],
    fixed_costs: List[float]
) -> List[List[int]]:
    """
    Solves the Capacitated Facility Location Problem.

    Args:
        facility_capacities: Max capacity for each facility (Size M).
        customer_demands: Demand values for each customer (Size N).
        assignment_costs: Matrix (M x N) where [i][j] is the cost
                          to serve customer j from facility i.
        fixed_costs: Setup cost for opening each facility.

    Returns:
        A list of assignment pairs or an allocation matrix.
    """
    pass
```

---

## C.3.2. CEVRPTW

**[Code Template: {{function_template}}]**

```python
def solve_cevrptw(
    distance_matrix: np.ndarray,
    demands: np.array,
    time_windows: np.ndarray,
    service_times: np.ndarray,
    vehicle_capacity: int,
    battery_capacity: float,
    station_indices: List[int]
) -> List[int]:
    """
    Solves the Capacitated Electric VRP with Time Windows.

    Args:
        distance_matrix: (N x N) distance between all nodes.
        demands: (N) Demand of nodes (0 for depot/stations).
        time_windows: (N x 2) [Earliest, Latest] arrival times.
        service_times: (N) Duration required at each node.
        vehicle_capacity: Max load per vehicle.
        battery_capacity: Max battery energy units.
        station_indices: Indices of nodes that are charging stations.

    Returns:
        Flattened list of routes (e.g., [0, 1, 5, 0, 0, 2, 0]).
        Must start/end with Depot (0).
    """
    pass
```

# D. Experimental Settings

## D.1. Experiments Dataset

To comprehensively evaluate the generalization capability of A₂DEPT, we employ a diverse suite of benchmarks ranging from standard NP-hard problems to complex, high-constraint real-world scenarios. The datasets are categorized into two groups: the **FrontierCO Benchmark** for standard performance evaluation, and specialized datasets for **High-Constraint Problems**. Detailed specifications are summarized in Table 5.

### D.1.1. STANDARD BENCHMARKS

As discussed in the main results, we select four representative problems from the FrontierCO (Feng et al., 2025): benchmark to cover graph-based, routing, location, and scheduling domains. For MIS problem, we derive instances from the 2nd DIMACS (Johnson & Trick, 1996) Challenge and BHOSLIB (Xu et al., 2007), covering both dense and sparse graph topologies. In the domain of routing, we address CVRP using the Golden and Arnold instances from CVRPLib (Reinelt, 1991), which represent classic logistics scenarios with varying customer distributions. For large-scale location decisions, we employ the CFLP sourced from Avella and Boccia (Avella & Boccia, 2009b; Avella et al., 2009), featuring instances with up to 1,000 facilities and customers. Finally, to test resource allocation under flexibility constraints, we adopt FJSP instances from the Behnke and Naderi dataset (Behnke & Geiger, 2012; Naderi & Roshanaei, 2021).

### D.1.2. HIGH-CONSTRAINT BENCHMARKS

To further test A₂DEPT's ability to handle complex constraints, we introduce two specialized datasets.

**ESOGU-CEVRPTW.** We utilize the ESOGU-CEVRPTW dataset (Aslan et al., 2025), a real-world benchmark generated for the Eskisehir Osmangazi University (ESOGU) Meselik Campus. This dataset introduces rigorous constraints including battery capacity, charging scheduling, and strict time windows. The base topology comprises 118 customers, 10 distinct

*Table 5.* **Summary of Dataset Specifications.** The table reports the source, instance count, and scale of all benchmarks.

| Problem | Source | Distribution / Topology | #Inst. | Scale |
|---------|--------|------------------------|--------|-------|
| *Standard Benchmarks* | | | | |
| **MIS** | DIMACS / BHOSLIB | Benchmark graphs (mixed) | 36 | $|V| \in [200, 1500]$ |
| **CVRP** | CVRPLib (Golden/Arnold) | Geometric (2D) | 20 | $n \in [200, 483]$ |
| **CFLP** | Avella & Boccia (Test Bed 1) | Large-scale bipartite | 20 | 1000 facilities, 1000 customers |
| **FJSP** | Behnke & Geiger | Flexible operations | 60 | Jobs $J \in [20, 100]$, Machines $M \in [20, 60]$ |
| *High-Constraint Benchmarks* | | | | |
| **CEVRPTW** | ESOGU-CEVRPTW | Real-world campus map | 45 | $n \in \{5, 10, 20, 40, 60\}$ |
| **MRCPSP** | PSPLIB (c15) | Precedence graphs | 551 | 16 jobs, 3 modes |

charging stations, and a single depot. To ensure diversity, the dataset contains 45 instances divided into 5 scale groups ($N \in \{5, 10, 20, 40, 60\}$ customers). Furthermore, each group includes three spatial distribution types—Random (R), Clustered (C), and Random-Clustered (RC)—and features varying time window widths (Wide, Medium, and Narrow) to modulate difficulty.

**PSPLIB.** For project scheduling with complex resource trade-offs, we employ the MRCPSP instances from the PSPLIB library (Sprecher & Kolisch, 1996). Specifically, we focus on the c15 dataset, which comprises 551 diverse instances. Unlike standard FJSP, these instances introduce a multi-modal decision layer, requiring the algorithm to simultaneously determine the sequence of activities and the execution mode for each task. These decisions are subject to strict limits on both renewable and non-renewable resources, significantly increasing the complexity of the solution space.

### D.2. Evaluation Dataset

In this section, we provide a more detailed introduction to the setup of evaluation datasets B used in the evaluation phase.

To foster the development of scale-invariant heuristics, we employ a multi-scale generation strategy for the evaluation dataset. Unlike approaches that optimize on a single fixed size, our framework generates synthetic instances across four distinct complexity tiers: Small, Medium, Large, and Extra Large. As detailed in Table 6, for each problem, we construct a fitness set $\mathcal{B}$ containing 16 synthetic instances, evenly distributed across the four tiers (4 instances per tier). This heterogeneous mix exposes the LLM to a wide spectrum of problem scales—ranging from trivial feasibility checks to large-scale optimization challenges—thereby encouraging the discovery of robust, generalized logic.

A significant challenge in multi-scale optimization is the magnitude disparity of objective values across problem sizes. Direct summation of raw objective values would introduce severe scale bias, causing the evolutionary search to disproportionately prioritize large-scale instances while neglecting improvements on smaller ones. To address this, we employ a scale-normalized scoring metric. We unify the optimization direction by treating all tasks as maximization problems. For a given instance $I$ with raw objective value $\mathcal{O}(I)$, the normalized fitness score $S(I)$ is defined as:

$$S(I) = \begin{cases} \frac{\mathcal{O}(I)}{\sigma(I)} & \text{for maximization tasks} \\ -\frac{\mathcal{O}(I)}{\sigma(I)} & \text{for minimization tasks} \\ -\infty & \text{if execution fails or solution is infeasible} \end{cases} \tag{47}$$

Here, $\sigma(I)$ represents the fundamental complexity dimension of the instance, serving as the normalization factor:

- **Routing & Location (CVRP, CEVRPTW, CFLP):** $\sigma(I) = N$ (number of customers), interpreting the metric as the average cost per customer.

- **Scheduling (FJSP, MRCPSP):** $\sigma(I) = J$ (number of jobs), representing the average processing time per job.

- **Graph (MIS):** $\sigma(I) = |V|$ (number of vertices), representing the node selection ratio.

*Table 6.* **Specifications of Multi-Scale Evolution Datasets.** Four complexity tiers used to build the evaluation dataset B, each scale tier contains 4 instances.

| Problem | Scale Tier | Dimensions | Key Constraints / Parameters |
|---|---|---|---|
| **CFLP** | Small | 25 Fac. $\times$ 25 Cust. | Cap $\sim U[5, 100]$, Demand $\sim U[5, 20]$, FixedCost $\sim U[100, 500]$ |
| | Medium | 50 Fac. $\times$ 50 Cust. | |
| | Large | 100 Fac. $\times$ 100 Cust. | |
| | Extra Large | 200 Fac. $\times$ 200 Cust. | |
| **CVRP** | Small | $N = 25$ Customers | Vehicle Capacity $\sim U[20, 40]$, Demand $\sim U[1, 10]$, 2D Euclidean Space |
| | Medium | $N = 50$ Customers | |
| | Large | $N = 100$ Customers | |
| | Extra Large | $N = 200$ Customers | |
| **FJSP** | Small | 10 Jobs $\times$ 5 Mach. | Ops per Job $\sim U[5, 15]$, Processing Time $\sim U[10, 100]$, Flexible Machine Eligibility |
| | Medium | 20 Jobs $\times$ 10 Mach. | |
| | Large | 50 Jobs $\times$ 20 Mach. | |
| | Extra Large | 100 Jobs $\times$ 50 Mach. | |
| **MIS** | Small | $V = 50$ Vertices | Edge Probability $p \in [0.1, 0.3]$, Random Graphs (Erdős-Rényi) |
| | Medium | $V = 100$ Vertices | |
| | Large | $V = 250$ Vertices | |
| | Extra Large | $V = 500$ Vertices | |
| **CEVRPTW** | Small | 20 Cust. + 3 Stn. | Battery=5.0, Horizon=20.0, Service Time=0.05, Relaxed Time Windows |
| | Medium | 40 Cust. + 4 Stn. | |
| | Large | 60 Cust. + 5 Stn. | |
| | Extra Large | 80 Cust. + 6 Stn. | |
| **MRCPSP** | Small | 10 Jobs | 3 Modes per Job, 2 Renewable / 2 Non-Renewable Resources, Budget Factor $\alpha = 0.8$ |
| | Medium | 20 Jobs | |
| | Large | 30 Jobs | |
| | Extra Large | 40 Jobs | |

Finally, the global fitness $S(P)$ for a candidate algorithm is calculated by aggregating instance-level scores over the fitness set $\mathcal{B}$:

$$S(P) = \frac{1}{|\mathcal{B}|} \sum_{I \in \mathcal{B}} S(I). \tag{48}$$

This normalization ensures that instances of all sizes contribute equally to the global fitness function, effectively mitigating scale bias.

## E. Extended Experimental Results

Due to space constraints in the main text, we present additional experimental analyses in this appendix.

### E.1. Sensitivity Analysis of LLM

To investigate the impact of the LLM's reasoning capability on A$_2$DEPT's performance, we conducted a comparative analysis using two representative model classes: the open-weight reasoning model DeepSeek-V3.2 and the proprietary lightweight model Gemini 2.5 Flash-lite. We benchmarked A$_2$DEPT against four established baselines—FunSearch (Romera-Paredes et al., 2024), EoH (Liu et al., 2024a), ReEvo (Liu et al., 2024a), and MCTS-AHD (Zheng et al., 2025)—across four problem domains. The detailed results are presented in Table 7. Two key observations emerge:

First, when empowered by the advanced reasoning capabilities of DeepSeek-V3.2, A$_2$DEPT achieves superior performance across all tasks. This indicates that A$_2$DEPT's structural search mechanism effectively amplifies the logical depth of high-capacity reasoning models. Second, upon transitioning to the lightweight Gemini, A$_2$DEPT exhibits a more pronounced performance degradation compared to evolutionary baselines. This disparity suggests that A$_2$DEPT relies heavily on

*Table 7.* **Impact of LLM Backends on AHD Performance.** Comparison of A₂DEPT using DeepSeek-V3.2 (Think Mode) vs. Gemini 2.5 Flash. The scientific notation multiplier is indicated in the column headers. Bold entries denote the best result within each backend group.

| Task | CFLP | | CVRP | | FJSP | | MIS | |
|---|---|---|---|---|---|---|---|---|
| **Metric** | Obj. $(\times 10^5)\uparrow$ | Gap | Obj. $(\times 10^5)\downarrow$ | Gap | Obj. $(\times 10^4)\downarrow$ | Gap | Obj. $(\times 10^3)\downarrow$ | Gap |
| *LLM: DeepSeek-V3.2 (Think Mode)* | | | | | | | | |
| FunSearch | 7.20 | 19.80% | 1.21 | 22.22% | 1.64 | 31.20% | 2.45 | 14.34% |
| EoH | 7.05 | 17.30% | 1.15 | 16.16% | 1.66 | 32.80% | 2.65 | 7.34% |
| ReEvo | 6.92 | 15.14% | 1.20 | 21.21% | 1.64 | 31.20% | 2.58 | 9.79% |
| MCTS-AHD | 6.78 | 12.81% | 1.16 | 17.17% | 1.61 | 28.80% | 2.52 | 11.89% |
| A₂DEPT (Ours) | **6.48** | **7.82%** | **1.05** | **8.54%** | **1.44** | **15.20%** | **2.79** | **2.45%** |
| *LLM: Gemini 2.5 flash-lite* | | | | | | | | |
| FunSearch | 7.55 | 25.62% | 1.26 | 27.27% | 1.67 | 33.60% | 2.34 | 18.18% |
| EoH | 7.25 | 20.63% | 1.14 | 15.15% | 1.65 | 32.00% | 2.48 | 13.29% |
| ReEvo | 7.15 | 18.97% | 1.23 | 24.24% | 1.62 | 29.20% | **2.66** | **7.13%** |
| MCTS-AHD | 7.38 | 22.79% | 1.17 | 18.18% | 1.66 | 32.80% | 2.45 | 14.34% |
| A₂DEPT (Ours) | **6.98** | **16.14%** | **1.12** | **13.13%** | **1.61** | **28.56%** | 2.65 | 7.41% |

*Table 8.* **Comparison with OPRO and LMEA on TSP.** The table reports the Optimality Gap (%) on random uniform (Rue) and clustered (Clu) instances.

| Type | Size ($N$) | LMEA | OPRO | A₂DEPT (Ours) |
|---|---|---|---|---|
| **Rue** (Uniform) | 10 | **0.00** | **0.00** | **0.00** |
| | 15 | **0.06** | 5.23 | 2.87 |
| | 20 | 3.94 | 26.30 | **2.78** |
| | 25 | 18.72 | 53.59 | **3.56** |
| **Clu** (Clustered) | 10 | **0.00** | **0.00** | 0.01 |
| | 15 | 0.11 | 8.13 | **0.06** |
| | 20 | 4.05 | 19.83 | **2.65** |
| | 25 | 10.06 | 48.25 | **3.46** |

the model's capacity to deduce logical structures. In contrast, evolutionary methods depend primarily on stochastic recombination; while this renders them less sensitive to reasoning depth, it inherently limits their potential for achieving structural breakthroughs.

### E.2. Comparison with LLM-as-Optimizer Paradigm

We further evaluate the proposed LLM-as-Programmer paradigm by comparing it with OPRO (Optimization by PROmpting) (Yang et al., 2024; Liu et al., 2024b), a representative LLM-as-Optimizer approach. Experiments are conducted on the EUC-2D TSP testbed with both uniform-random (Rue) and clustered (Clu) instance families generated using the DIMACS (Gutin & Punnen, 2006) generators. We compare A₂DEPT with OPRO and LMEA across four problem scales ($N \in 10, 15, 20, 25$).

Table 8 summarizes the results. OPRO performs well on the smallest instances but degrades as $N$ increases, illustrating a context scalability limitation: directly prompting an LLM for complete solutions becomes brittle as the combinatorial search space grows within a fixed context window. LMEA shows a similar sensitivity to scale.

In contrast, A₂DEPT exhibits stronger zero-shot scaling. Because A₂DEPT evolves a solver that can be applied across instance sizes—rather than producing a single solution for each instance—it maintains stable performance as the scale increases. Overall, these findings support the view that synthesizing algorithmic logic offers better generalization and robustness than one-shot solution prompting.

*Table 9.* **Performance Comparison at Different Convergence Stages.** Optimality gap (%) at $T = 200$ (Early Stage) and $T = 1000$ (Deep Search) using DeepSeek-V3.2. Bold entries denote the best result in each column.

| Method | Stage I: Early Search ($T = 200$) | | | | Stage II: Deep Search ($T = 1000$) | | | |
|---|---|---|---|---|---|---|---|---|
| | CFLP | CVRP | FJSP | MIS | CFLP | CVRP | FJSP | MIS |
| FunSearch | 24.87% | 25.23% | 29.47% | 19.38% | 15.73% | 18.89% | 22.12% | 14.72% |
| EoH | 20.09% | 19.27% | 31.11% | 15.32% | 13.82% | 16.47% | 23.10% | 12.33% |
| ReEvo | 19.45% | 26.23% | 30.58% | 15.33% | 14.90% | 13.67% | 25.98% | 11.86% |
| MCTS-AHD | **17.67%** | 21.29% | 29.52% | 18.21% | 11.34% | 15.42% | 24.51% | 12.35% |
| A₂DEPT (Ours) | 21.87% | **14.53%** | **22.38%** | **12.98%** | **9.27%** | **8.51%** | **9.95%** | **4.16%** |

### E.3. Performance at Different Convergence Stages

To further assess search efficiency and long-horizon evolutionary potential, we report results at two budgets: an early stage ($T$=200) and a deep search stage ($T$=1000). All methods use the same backbone model (**DeepSeek-V3.2**) to ensure a fair comparison across search mechanisms. The optimality gaps are summarized in Table 9.

At the early stage, A₂DEPT achieves strong performance on most benchmarks, indicating that its hierarchical operators can quickly surface promising structural hypotheses and translate them into executable improvements. On CFLP, however, A₂DEPT is comparatively less competitive under a limited budget, reflecting a deliberate exploration bias: the top-down operators spend early samples on restructuring the overall workflow rather than aggressively tuning within a fixed template.

With an extended budget, A₂DEPT exhibits substantially stronger evolutionary potential than competing AHD frameworks. While population-based baselines and MCTS-AHD typically show diminishing returns as the search progresses, A₂DEPT continues to improve consistently, eventually achieving the best results across all tasks. This trend suggests that open-ended system-level redesign can overcome the expressiveness ceiling of template-bound component tuning, and that investing early budget into structural exploration yields superior long-horizon gains.

To complement the two-point snapshots in Table 9, Figure 3 visualizes the full convergence trajectory on CVRP as best-so-far gap against the number of evaluated programs. A₂DEPT shows a sharp initial drop during the first few hundred evaluations, driven by macro-mutation discovering paradigm-level structural changes (e.g., the constructive-to-ILS transition in Appendix F.1), followed by sustained fine-grained improvement. In contrast, the baselines enter their late-phase plateau earlier, reflecting the ceiling imposed by template-bound heuristic tuning.

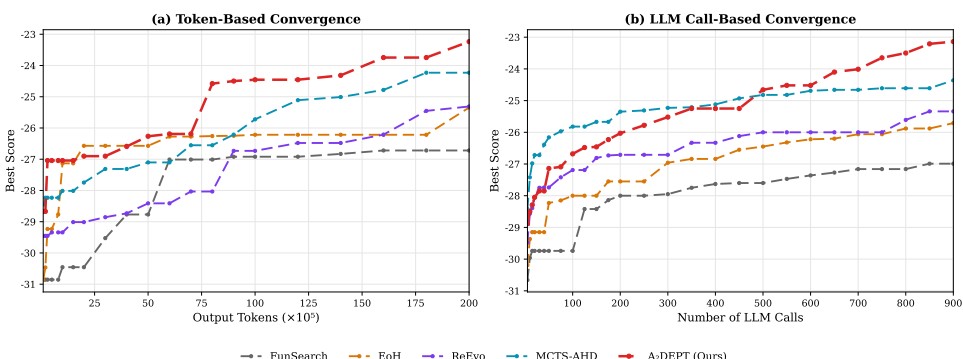

*Figure 3.* **Convergence trajectory on CVRP.** Best-so-far optimality gap (%) as a function of the number of evaluated programs. A₂DEPT exhibits a sharp initial drop driven by macro-mutation-level structural breakthroughs, followed by sustained fine-grained improvement, whereas the baselines plateau earlier.

### E.4. LLM Scaling and Token-Budget Analysis

To further probe A₂DEPT's reliance on the backbone model and its efficiency relative to LLM-call cost, we conduct two additional analyses on top of Appendix E.1: (i) a controlled scaling study across Qwen-3.5 variants ranging from 4B to 397B parameters, and (ii) a token-budget-controlled comparison that equalizes monetary cost across methods.

*Table 10.* **Normalized scores on CVRP across different model scales (Qwen-3.5).** Higher is better ↑. Bold: best result in each column.

| Method | 4B | 9B | 27B | 35B | 122B | 397B |
|---|---|---|---|---|---|---|
| EoH | 0.673 | 0.693 | 0.752 | 0.785 | 0.819 | 0.831 |
| MCTS-AHD | 0.694 | 0.739 | 0.777 | 0.792 | 0.805 | 0.846 |
| A$_2$DEPT (Ours) | **0.756** | **0.806** | **0.889** | **0.893** | **0.919** | **0.933** |

*Table 11.* **Normalized scores under equalized token budgets.** Comparison of A$_2$DEPT against baselines under fixed total token-cost budgets on DeepSeek v3.2 (Non-Think Mode). Higher is better ↑. Bold: best result in each column.

| Method | 10 CNY | | | | 5 CNY | | | |
|---|---|---|---|---|---|---|---|---|
| | CFLP | CVRP | FJSP | MIS | CFLP | CVRP | FJSP | MIS |
| A$_2$DEPT (Ours) | **0.928** | **0.947** | **0.857** | **0.960** | **0.893** | **0.915** | **0.839** | **0.938** |
| FunSearch | 0.851 | 0.841 | 0.749 | 0.862 | 0.816 | 0.817 | 0.735 | 0.839 |
| EoH | 0.868 | 0.862 | 0.787 | 0.896 | 0.846 | 0.825 | 0.725 | 0.871 |
| ReEvo | 0.875 | 0.862 | 0.769 | 0.886 | 0.851 | 0.841 | 0.737 | 0.864 |
| MCTS-AHD | 0.885 | 0.853 | 0.793 | 0.890 | 0.847 | 0.837 | 0.748 | 0.853 |

**LLM scaling study (Qwen-3.5, 4B–397B).** A natural concern is whether the gains reported in the main paper are mainly attributable to using a strong reasoning model. To isolate this factor, we fix the task to CVRP and run all methods with the same family of backbones (Qwen-3.5) at six parameter scales, keeping search hyperparameters identical to the main experiments. Results are summarized in Table 10. A$_2$DEPT outperforms EoH and MCTS-AHD at *every* scale, including the smallest 4B model, and its scaling slope is steeper than either baseline, indicating that A$_2$DEPT more effectively converts additional LLM capability into optimization performance. The degradation observed on Gemini 2.5 Flash-lite in Table 7 is consistent with this picture: A$_2$DEPT benefits from but does not hinge on a specific model, though it does require a minimum level of code-generation ability because macro-mutation and dependency repair both rely on semantically coherent generation.

**Equalized token-budget comparison.** The main experiments fix the number of LLM calls rather than the total token cost, which could in principle give A$_2$DEPT an advantage if it consumes more tokens per call. To rule out this concern, we additionally run all methods on CFLP, CVRP, FJSP, and MIS under two fixed monetary budgets—approximately 5 CNY and 10 CNY of DeepSeek v3.2 (Non-Think Mode) API usage ($\approx$\$0.7 and \$1.4 USD, respectively)—stopping each method once its cumulative cost reaches the cap. Scores are normalized to $[0, 1]$ per task for cross-benchmark comparison. As shown in Table 11, A$_2$DEPT remains the strongest method under both budgets on all four benchmarks, confirming that its advantage is not an artifact of consuming more tokens. When the budget is halved from 10 to 5 CNY, all methods degrade, but A$_2$DEPT continues to lead, indicating that the gains come from the search framework itself rather than from elevated token consumption. Jointly with Table 16, these results show that A$_2$DEPT achieves its improved search effectiveness at comparable or lower token cost relative to MCTS-AHD, while avoiding the token inflation observed when other baselines are moved to the AAD setting.

### E.5. Parameter Sensitivity Analysis

To assess the robustness of A$_2$DEPT to hyperparameter choices across domains, we vary three dynamic-control parameters: the stagnation tolerance ($N_{stall}$), the temperature decay factor ($\alpha$), and the penalty coefficient ($\lambda$). Experiments are conducted on the CVRP and FJSP validation sets with a fixed budget of $T=500$. Each setting is repeated for 3 runs with different random seeds, and we report the average optimality gap in Table 12. Overall, A$_2$DEPT is stable within a reasonable range of values, and we use the highlighted defaults in all experiments.

**Stagnation Tolerance ($N_{stall}$).** $N_{stall}$ controls when re-annealing is triggered after a period without improvement. Table 12 shows a non-monotonic trade-off: overly aggressive re-annealing can disrupt local refinement, whereas overly conservative settings delay escaping local optima. The default $N_{stall}=3$ provides a robust balance across both routing and scheduling tasks, while slightly larger values can be preferable on harder scheduling instances.

**Decay Factor ($\alpha$).** $\alpha$ determines the cooling rate of the simulated annealing schedule. Very fast cooling tends to lock the search into early choices, while very slow cooling can allocate too much budget to exploration and slow convergence within the fixed evaluation budget. In our experiments, intermediate values (around 0.90–0.95) perform consistently well, and we adopt $\alpha=0.95$ as a robust default across tasks.

**Penalty Coefficient ($\lambda$).** $\lambda$ scales the negative feedback assigned to operators that produce invalid or low-quality code. Small penalties provide a weak learning signal, whereas overly strict penalties can make the scheduler prematurely abandon exploratory operators after occasional failures. A moderate penalty (our default $\lambda=0.8$) yields stable behavior across both benchmarks.

*Table 12.* **Impact of Dynamic Control Parameters.** Comparison of optimality gaps on CVRP and FJSP. The default configuration (highlighted) is used throughout the paper.

| Parameter | Value | Optimality Gap (%) | |
| --- | --- | --- | --- |
| | | CVRP | FJSP |
| **Stagnation Tolerance** ($N_{\text{stall}}$) | 1 | 17.35% | 31.77% |
| | **3** | **8.79%** | 18.95% |
| | 5 | 9.86% | **18.78%** |
| | 7 | 10.59% | 23.92% |
| **Decay Factor** ($\alpha$) | 0.80 | 9.72% | 21.21% |
| | 0.90 | **8.75%** | 19.47% |
| | **0.95** | 8.79% | **18.95%** |
| | 0.98 | 10.32% | 20.25% |
| **Penalty Coefficient** ($\lambda$) | 0.5 | 8.99% | 26.21% |
| | **0.8** | **8.79%** | **18.95%** |
| | 1 | 9.52% | 23.50% |

### E.6. Sensitivity to Evaluation Wall-Clock Budget

The main experiments cap each generated solver program at a wall-clock limit of 120 seconds to solve the full evaluation dataset, balancing fairness across paradigms with manageable search cost. A natural concern is whether this cap may systematically favor or disfavor certain algorithmic styles (e.g., constructive vs. iterative-improvement methods). To assess this, we re-run A$_2$DEPT and the strongest evolutionary baseline (EoH) on CVRP under six time budgets from 30 to 300 seconds, keeping every other setting identical to the main experiments. Per-benchmark normalized scores are reported in Table 13.

*Table 13.* **Normalized scores on CVRP under different per-instance wall-clock budgets.** Higher is better. Both methods use DeepSeek v3.2 (Non-Think Mode). Our paper uses the 120 s setting, shaded for reference.

| Time (s) | 30 | 60 | 120 | 180 | 240 | 300 |
| --- | --- | --- | --- | --- | --- | --- |
| A$_2$DEPT (Ours) | 0.905 | 0.907 | **0.914** | 0.919 | 0.928 | 0.925 |
| EoH | 0.792 | 0.811 | 0.819 | 0.824 | 0.823 | 0.823 |

Both methods monotonically benefit from larger budgets up to around 180 s; beyond that, gains taper off for EoH and show only marginal improvement for A$_2$DEPT. Between 120 s and 180 s the scores are close (within $\approx 0.005$ for A$_2$DEPT, $\approx 0.005$ for EoH), indicating that the ranking of methods is stable in this regime. We therefore view the 120 s setting as a practical and balanced operating point: it is large enough to avoid truncation artifacts on either side, and small enough to keep the overall search affordable across all six benchmarks.

### E.7. Source of the GLS Advantage on FJSP

The main text (Section 4.1.2) observes that GLS outperforms all AAD-style methods on FJSP. To pinpoint the source of this advantage, we replace the LLM-designed guidance matrix with a matrix sampled uniformly at random (GLS$_{\text{rand}}$) while

keeping the fixed expert-designed neighborhood operators unchanged. Table 14 reports the normalized scores (higher is better).

*Table 14.* **GLS with LLM-designed vs. random guidance.** Normalized scores under identical fixed operators. Higher is better.

| Task | GLS$_{LLM}$ | GLS$_{rand}$ |
|------|------|------|
| CFLP | 0.621 | 0.302 |
| CVRP | 0.845 | 0.830 |
| FJSP | 0.873 | 0.808 |
| MIS | 0.844 | 0.818 |

The contrast between tasks is revealing. On FJSP, replacing LLM guidance with a random matrix causes only a modest drop ($0.873 \rightarrow 0.808$), which means the fixed expert-designed local-search operators alone already deliver most of the performance. On CFLP, in contrast, the drop is dramatic ($0.621 \rightarrow 0.302$), indicating that the operator set alone is insufficient and LLM-designed knowledge matters substantially. Hence, the GLS lead on FJSP should be attributed primarily to the inductive bias of its hand-engineered neighborhood operators rather than to LLM-designed knowledge. This makes template-bound AHD and open-ended AAD *complementary*: AHD is advantageous when an effective operator set is available, while AAD is preferable when no such operators exist and the solver workflow itself must be discovered.

### E.8. Out-of-Distribution Generalization

The main experiments evaluate A$_2$DEPT on each benchmark under a fixed instance distribution. A natural question is whether the solvers discovered by A$_2$DEPT overfit to the distribution they were evolved on, or whether they learn transferable algorithmic strategies. To probe this, we use the VRPTW Solomon benchmark, which provides three canonical spatial distributions: clustered (C), random (R), and mixed random-clustered (RC). We evolve a separate solver with A$_2$DEPT on each of these three distributions, then evaluate each resulting solver across all three distributions, yielding a $3 \times 3$ transfer matrix.

*Table 15.* **Cross-distribution generalization on VRPTW (Solomon benchmark).** Each row is a solver evolved on one distribution; each column is the test distribution. Entries are mean Gap (%); lower is better.

| Evolve\Test | C | R | RC |
|------|------|------|------|
| C | 23.05 | 32.52 | 28.77 |
| R | 24.37 | 28.40 | 26.07 |
| RC | 25.87 | 31.07 | 28.50 |

Two observations emerge from Table 15. First, no solver collapses when tested out-of-distribution: performance degrades gracefully rather than catastrophically, with cross-distribution gaps staying within roughly $1.5\times$ of the within-distribution gap across all cells. Second, the solver evolved on the R (random uniform) distribution achieves the best *row-averaged* performance across the three test distributions, indicating the strongest overall cross-distribution robustness. This suggests that A$_2$DEPT does not merely fit surface statistics of the training distribution but tends to discover generalizable strategies—consistent with the fact that it evolves complete solver logic rather than distribution-specific parameters.

### E.9. Resource Consumption Analysis

Table 16 summarizes wall-clock time and input/output token usage (in millions) for *designing* heuristics/solvers across four domains. Following the reporting protocol of MCTS-AHD (Zheng et al., 2025), we report both runtime and token consumption under the step-by-step constructive AHD template, and additionally include open-ended AAD variants for EoH and ReEvo.

Under the AHD template, A$_2$DEPT remains more efficient than MCTS-AHD in both time (e.g., 2.5–3.0h vs. 3.0–3.5h) and tokens (0.6–0.7M input vs. 1.0–1.4M). This advantage is consistent with our design choices: hybrid selection reduces unnecessary expansions, while program maintenance mitigates repeated failures due to non-executable generations. Compared with evolutionary AHD baselines (FunSearch/EoH/ReEvo), A$_2$DEPT incurs higher wall-clock time due to dependency analysis and iterative repair, but its token usage stays in a comparable range.

*Table 16.* **Analysis of Computational Resources.** Wall-clock time (hours) and token usage (millions) for designing heuristics/solvers.

| Method / Framework | Metric | CFLP | CVRP | FJSP | MIS |
|---|---|---|---|---|---|
| *Framework: Step-by-step Construction* | | | | | |
| FunSearch | Time (h) | 1.0 | 1.0 | 1.0 | 1.0 |
| | Input (M) | 0.4 | 0.4 | 0.5 | 0.4 |
| | Output (M) | 0.3 | 0.3 | 0.3 | 0.3 |
| EoH | Time (h) | 1.5 | 1.5 | 2.0 | 1.5 |
| | Input (M) | 0.5 | 0.4 | 0.6 | 0.4 |
| | Output (M) | 0.3 | 0.3 | 0.4 | 0.3 |
| ReEvo | Time (h) | 2.0 | 2.0 | 2.5 | 2.0 |
| | Input (M) | 0.8 | 0.8 | 1.0 | 0.7 |
| | Output (M) | 0.5 | 0.4 | 0.6 | 0.3 |
| MCTS-AHD | Time (h) | 3.5 | 3.0 | 3.0 | 3.5 |
| | Input (M) | 1.4 | 1.2 | 1.4 | 1.0 |
| | Output (M) | 0.6 | 0.5 | 0.5 | 0.3 |
| *Framework: AAD* | | | | | |
| EoH | Time (h) | 1.8 | 1.7 | 2.1 | 1.6 |
| | Input (M) | 1.5 | 1.6 | 1.2 | 0.9 |
| | Output (M) | 1.0 | 1.1 | 0.9 | 0.5 |
| ReEvo | Time (h) | 2.1 | 2.1 | 2.6 | 2.0 |
| | Input (M) | 1.6 | 1.9 | 1.3 | 1.0 |
| | Output (M) | 1.0 | 1.1 | 0.8 | 0.5 |
| **A₂DEPT (Ours)** | Time (h) | 3.0 | 2.5 | 2.5 | 2.5 |
| | Input (M) | 0.6 | 0.7 | 0.6 | 0.6 |
| | Output (M) | 0.5 | 0.5 | 0.4 | 0.4 |

Moving from template-bound AHD to open-ended AAD substantially increases token consumption for both EoH and ReEvo (roughly 0.9–1.9M input and 0.5–1.1M output), indicating that greater synthesis freedom can amplify repair and revision overhead even when runtime increases are modest. Overall, A₂DEPT achieves improved search effectiveness with moderate and controllable computational cost, while avoiding the token inflation observed in open-ended AAD baselines.

## F. Qualitative Analysis: Case Studies

In this section, we provide a qualitative analysis of representative success and observed failure modes in A₂DEPT. We first present a successful lineage on MIS to illustrate how our mechanisms jointly enable structural breakthroughs. We then analyze recurring failure modes that arise when moving from template-bound heuristic design to open-ended algorithm synthesis.

### F.1. Case Study I: Evolutionary Success and Paradigm Shift

This case study tracks the lineage of the top-performing algorithm on the MIS task. It provides qualitative evidence for the *interplay* of three key mechanisms: (1) **Boltzmann Supplementary Selection** for preserving diversity, (2) **Hierarchical Operators** for structural innovation, and (3) **Program Maintenance** for ensuring executability.

**Phase 1: Constructive Greedy Baseline (ID 2, Score 24.38).** The lineage starts from a standard enhanced greedy heuristic. As shown in Figure 4(a), ID 2 constructs an independent set by iteratively selecting a node with minimum degree and removing its neighbors. While computationally cheap, this purely constructive paradigm offers limited revision capability.

**Phase 2: Preserving Diversity (ID 19, Score 25.06).** In Generation 3, the system produced ID 19, which combined signals from two parent heuristics. Despite this structural variation, ID 19 was initially rejected by the primary Simulated Annealing pressure due to insufficient score improvement. However, our *Boltzmann Supplementary Selection* re-introduced

it into the parent pool, preserving a pivotal evolutionary pathway that would otherwise have been pruned.

**Phase 3: Macro-level Paradigm Shift (ID 70, Score 25.94).** Using the retained lineage, Generation 4 (ID 70) exhibits a massive structural jump. As illustrated in Figure 4(b), the LLM moves beyond constructive greedy logic and synthesizes a complete ILS framework. This includes a `kick_move` for perturbation, a multi-operator `local_search_step`, and basin-hopping control logic. This transition confirms the ability of our hierarchical operators to perform non-local program rewrites.

**Phase 4: Efficiency Refinement (ID 318, Score 26.44).** In later generations (e.g., Generation 24), the focus shifted to fine-grained refinement. ID 318 introduced priority queues (Heaps) via `heapq` to support repeated ranking operations efficiently (Figure 4(c)). Throughout this process, the Program Maintenance mechanism was critical in detecting and repairing broken dependencies as the code grew into a complex multi-module system.

```
# ID 2: Simple Greedy Construction
def solve_mis(adjacency_matrix, n_nodes):
    candidate_nodes = np.arange(n_nodes)
    while len(candidate_nodes) > 0:
        next_node = select_next_node(...)    # min-degree style rule
        independent_set.append(next_node)
        # ... remove neighbors ...
    return independent_set
```

*(a)* Generation 1: Baseline Constructive Heuristic

```
# ID 70: Paradigm Shift to Iterated Local Search (ILS)
def solve_mis(adjacency_matrix, n_nodes):
    current_set = greedy_initial_solution(...)
    for iteration in range(max_iterations):
        if no_improve_count >= max_no_improve:
            current_set = kick_move(current_set, ...)    # perturbation
        new_set = local_search_step(current_set, ...)    # add/swap/plateau
        if len(new_set) > len(best_set):
            best_set = new_set.copy()
    return sorted(best_set)
```

*(b)* Generation 4: Structural Breakthrough Enabled by Hierarchical Operators

```
# ID 318: Heuristic Guidance + Data Structure Optimization
import heapq  # efficiency: priority queue operations

def local_search_step(current_set, ...):
    if add_candidates:
        degrees = [np.sum(adj[node]) for node in add_candidates]
        min_deg = min(degrees)
        best = [n for n, d in zip(add_candidates, degrees) if d == min_deg]
        node_to_add = random.choice(best)  # heuristic-guided choice
        current_set.add(node_to_add)
        return current_set
```

*(c)* Generation 24: Efficiency Refinement via Heaps and Heuristic Guidance

*Figure 4.* **Evolutionary trajectory on MIS.** The solver evolves from a constructive greedy heuristic (ID 2) to an ILS-based framework (ID 70) through non-local structural edits, and later improves efficiency and decision quality (ID 318).

### F.2. Case Study II: Failure Analysis and AAD Risks

To provide a comprehensive view of open-ended AAD, we analyze recurring failure modes observed on MIS. One prominent failure mode is **incomplete system-level refactoring**: the LLM initiates a global change of data representation, but fails to consistently enforce the new contract across modules. Unlike AHD, where data structures are typically fixed, A$_2$DEPT allows the LLM to redefine the solver's data representation. While this enables profound innovations (like the heap optimization in Phase 4 of Appendix F.1), it also opens the door to catastrophic structural inconsistencies.

**The Bitmask Refactoring Trap (IDs 158, 160).** We analyzed a cluster of candidates that are marked as non-executable due to runtime errors. In these instances, the LLM attempted a theoretically sound optimization: replacing standard Python `Set` operations with bitmasking (using integers to represent sets of nodes) . The model correctly identified that bitwise

operations could offer $O(1)$ complexity for set intersections on dense graphs and successfully implemented helper functions like `set_to_bitmask`.

**The Failure Mechanism: Partial Migration.** The crash occurred because the refactoring was *partial*. As shown in Figure 5, while the mutation operators allowed the LLM to rewrite the core search logic to use bitmasks, the cognitive load of maintaining this new "Bitmask Protocol" across all system modules proved too high. Notably, this is not a missing-dependency issue: the program is syntactically complete, yet violates a cross-module data contract, which is harder to recover from with dependency closure alone. Specifically, the LLM failed to enforce the new data contract in the perturbation operator (`kick_move`) and the final improvement phase, leading to **Type Mismatches** where bitmasks (integers) were passed to functions expecting iterables (sets).

**Insight.** This failure mode highlights a key trade-off: to enable system-level discovery, AAD must tolerate the instability arising from imperfect architectural migrations. It suggests that future work could benefit from static analysis tools or "type-aware" prompt augmentation to help the LLM maintain global consistency during radical data structure changes.

```
# ID 158: Partial Refactoring Failure (Bitmask vs Set)

# 1. The LLM correctly implements bitmask helpers (Good Idea)
def set_to_bitmask(node_set, n_nodes):
    mask = 0
    for node in node_set: mask |= 1 << node
    return mask

# 2. The Local Search is updated to use bitmasks (Complex Logic)
def improved_local_search(current_set, ...):
    # ... logic using bitwise OR/AND ...
    current_mask = set_to_bitmask(current_set, n_nodes)
    # LLM forgets to convert back to Set before returning!
    return current_mask  # <--- Returns INT (Bitmask)

# 3. The Main Loop or Next Operator crashes (Type Mismatch)
def solve_mis(adjacency_matrix, n_nodes):
    # ...
    # improved_local_search returns an INT, but current_set expects a SET
    current_set = improved_local_search(current_set, ...)

    # CRASH: kick_move expects a Set to calculate len(), but gets an Int
    if no_improve_count >= max_no_improve:
        # TypeError: object of type 'int' has no len()
        current_set = improved_kick_move(current_set, ...)
```

*Figure 5.* **Anatomy of a Failure (ID 158).** The LLM attempts to optimize the solver by introducing bitmasks. However, the refactoring is incomplete: `improved_local_search` returns a bitmask (int) instead of a set, causing a type error in the subsequent `kick_move` operator which expects a standard set. This illustrates the risk of "Type Consistency" loss during open-ended architectural redesign.

# G. Generalization to Continuous Optimization

To test whether A$_2$DEPT generalizes beyond discrete combinatorial optimization, we further evaluate it on three continuous domains: numerical ODE integration, root-finding, and feedback control. Together, these tasks assess whether the algorithm-design logic learned in discrete search can transfer to continuous mathematical settings.

## G.1. Discovery of Numerical ODE Solvers

To evaluate the capability of A$_2$DEPT in discovering numerical algorithms for continuous dynamics, we designed a rigorous benchmarking framework covering distinct numerical challenges.

**Experimental Setup.** The experiment dataset consists of four problem families commonly used in numerical ODE benchmarking (Hairer et al., 1993): (1) *Non-Stiff* (Harmonic Oscillator), (2) *Medium Non-Linear* (Van der Pol Oscillator), (3) *Stiff* (Robertson Kinetics), and (4) *High-Dimensional* (Discretized Heat Equation, $N = 20$). The evaluation metric is defined as $S = -\log_{10}(\text{Error}) - \lambda \log_{10}(\text{Cost})$, which balances accuracy and computational effort (function evaluations). Crucially, to rigorously test stability, we imposed a challenging fixed step size of $\Delta t = 0.05$ for all fixed-step solvers. This step size is significantly larger than the stability limit of classical explicit methods for the Stiff and High-Dimensional problems, thereby forcing the algorithm to discover implicit-like or highly stable mechanisms to ensure convergence. For adaptive solvers, we set a high precision tolerance ($rtol = 10^{-5}$) to simulate industrial usage standards.

*Table 17.* **ODE Solver Performance Comparison.** Scores reflect $-\log_{10}(\text{Error}) - \lambda\log_{10}(\text{Cost})$. Higher is better. "-20" indicates divergence. A₂DEPT achieves state-of-the-art performance on Stiff and High-Dimensional problems, outperforming specialized solvers like LSODA, though it struggles with medium non-linearity.

| Problem Family | Baselines | | | | | | Ours |
| --- | --- | --- | --- | --- | --- | --- | --- |
| | Euler | BwdEuler | RK4 | RK45 | LSODA | DOP853 | A₂DEPT |
| **1. Non-Stiff (Harmonic)** | 0.08 | 0.14 | **5.70** | 4.10 | 3.86 | 4.78 | 4.60 |
| **2. Medium (Van der Pol)** | -0.57 | -0.68 | 3.30 | 4.08 | 3.24 | **4.82** | -1.03 |
| **3. Stiff (Robertson)** | -20.00 | 3.42 | -20.00 | 3.21 | 4.73 | 1.75 | **5.56** |
| **4. High-Dim (Heat Eq)** | -3.45 | 1.20 | -20.00 | 5.30 | 5.04 | 5.32 | **5.99** |

**Baselines.** We benchmarked the evolved solver against a comprehensive suite of classical numerical integrators, organized into three distinct categories. First, we employed explicit fixed-step methods (Euler, RK4) (Hairer et al., 1993) to represent standard low-order and high-order approaches, which are anticipated to fail in stiff regimes given the large time step ($\Delta t$). Second, we included the implicit fixed-step method (Backward Euler) to benchmark stability; while inherently robust against stiffness, this approach typically incurs high computational costs due to the necessity of iterative root-finding. Finally, we compared against adaptive step-size methods (RK45, DOP853, LSODA) (Fehlberg, 1969; Dormand & Prince, 1980; Hindmarsh, 1983), representing industrial solvers. Among these, LSODA serves as the strongest baseline for stiff problems due to its capability to automatically switch between stiff (BDF) and non-stiff (Adams) integration schemes.

**Results Analysis.** Table 17 shows that A₂DEPT performs particularly well on stiff and high-dimensional problems, where standard explicit solvers diverge under the large fixed step size. It even surpasses specialized industrial solvers such as LSODA on the stiff benchmark, suggesting that the evolved method acquires unusually strong stability properties. On medium nonlinear dynamics, however, A₂DEPT underperforms adaptive methods, indicating that the current strategy lacks sufficiently fine-grained error-control logic in regimes that alternate between stiff and non-stiff behavior.

### G.2. Discovery of Numerical Root-Finding Algorithms

To evaluate A₂DEPT's capability in discovering optimization algorithms for continuous functions, we constructed a comprehensive benchmark covering diverse root-finding challenges.

**Experimental Setup.** We generated a dataset consisting of 400 stochastic instances across four distinct problem families. These include *Smooth Polynomials* representing well-behaved differentiable functions, *Ill-Conditioned Functions* characterized by multiple roots with vanishing derivatives , *Oscillatory Functions* featuring high non-linearity and local extrema, and *High-Dimensional Systems* ($N = 2$) requiring vector-valued solutions. To comprehensively assess algorithmic quality, we defined an efficiency-aware unified score $S = I_{\text{succ}} \cdot (1 - \alpha\ln(1 + N_{\text{fev}}) - \beta\ln(1 + N_{\text{iter}})) - (1 - I_{\text{succ}}) \cdot \gamma$, where $I_{\text{succ}}$ denotes the success indicator, and $N_{\text{fev}}$ and $N_{\text{iter}}$ represent the number of function evaluations and iterations, respectively. This metric explicitly rewards algorithms that converge successfully with minimal computational cost.

**Baselines.** We tested A₂DEPT against a wide spectrum of classical numerical solvers to ensure a rigorous comparison. For univariate problems, the baseline suite includes robust bracketing methods such as Bisection and Brent (Brent, 2013), which guarantee convergence given a valid interval; derivative-based methods like Newton-Raphson and Secant, which exploit gradient information for rapid convergence (Press et al., 2007); and acceleration techniques such as Steffensen's method (Steffensen, 1933). For multivariate systems, we employ specialized quasi-Newton solvers, specifically Broyden's method (Broyden, 1965) and Anderson acceleration (Anderson, 1965), which are designed to handle coupled non-linear equations without full Jacobian computation.

**Results Analysis.** The comparative results presented in Table 18 illustrate A₂DEPT's versatile performance profile. On well-behaved *Smooth Polynomials* and *Oscillatory* functions, A₂DEPT effectively matches the theoretical optimality of Newton-Raphson while significantly surpassing derivative-free methods, suggesting the rediscovery of gradient-exploitation logic. In the challenging *High-Dimensional* setting, A₂DEPT demonstrates superior generalization, substantially outperforming standard multivariate solvers like Broyden and Anderson that struggle with coupled non-linearities. Although A₂DEPT falls short of specialized bracketing methods like Brent on *Ill-Conditioned* problems due to the lack of strict

*Table 18.* **Performance Comparison on Numerical Root-Finding Tasks.** We report the average unified score (higher is better) over 400 stochastic instances. "–" indicates the method is not applicable to the problem dimension. A$_2$DEPT achieves the best or near-best performance across all categories, demonstrating superior generalization compared to specialized baselines.

| Task Family | Baselines | | | | | | | Ours |
| --- | --- | --- | --- | --- | --- | --- | --- | --- |
| | Bisect. | Brent | Newton | Secant | Steff. | Broyden | And. | A$_2$DEPT |
| 1. Smooth Poly. | 0.36 | 0.50 | **0.64** | 0.59 | 0.40 | – | – | 0.63 |
| 2. Ill-Cond. (Roots) | 0.65 | **0.75** | 0.30 | 0.23 | 0.45 | – | – | 0.51 |
| 3. Oscillatory | -1.15 | -1.20 | **0.66** | 0.59 | 0.64 | – | – | **0.66** |
| 4. High-Dim ($N = 2$) | – | – | – | – | – | 0.38 | 0.38 | **0.60** |

*Table 19.* **Long-Horizon Control Performance (T=5000).** Average stability scores over 5 random seeds (higher is better). "Fail" indicates divergence. The extended horizon highlights steady-state precision. A$_2$DEPT achieves SOTA results across all tasks, notably outperforming the expert-designed Energy Shaping on Pendulum and matching the theoretical optimal LQR on CartPole and Double Integrator.

| System Dynamics | Baselines | | | | | Ours |
| --- | --- | --- | --- | --- | --- | --- |
| | PID | SMC | Bang-Bang | Energy Shaping | LQR | A$_2$DEPT |
| 1. Pendulum Swing-Up | -1086.69 | -1086.69 | -1086.69 | -8.48 | -1086.69 | **-7.85** |
| 2. CartPole Stabilization | -22299.76 | -25683.72 | -4881.07 | – | -0.09 | **-0.08** |
| 3. Double Integrator | -2.18 | -2.17 | -7.74 | – | **-2.12** | -2.13 |

interval safeguards, it still robustly outperforms pure gradient-based methods that suffer from vanishing derivatives. In essence, A$_2$DEPT acts as a robust generalist, striking a favorable balance between the convergence speed of gradient methods and the stability of heuristic approaches across diverse topological landscapes.

### G.3. Discovery of Feedback Control Policies

To evaluate A$_2$DEPT's ability in discovering continuous feedback control laws, we constructed a benchmark consisting of three classic control tasks with distinct dynamic properties.

**Experimental Setup.** The evaluation includes three physical systems: Pendulum Swing-Up (Åström & Furuta, 2000) (underactuated non-linear), CartPole Stabilization (Barto et al., 2012) (multivariable unstable), and Double Integrator (linear canonical). All systems are simulated via RK4 integration (Press et al., 2007) with a time step of $\Delta t = 0.02s$. Crucially, we use an extended horizon of $T = 5000$ steps to evaluate long-horizon stability and steady-state accuracy. Performance is measured by a composite cost that heavily penalizes divergence ($\mathbb{I}_{\text{fail}}$), while also minimizing tracking error (MSE) and control effort.

**Baselines.** We compare against a diverse set of representative strategies. For linear regimes, we include PID (Åström & Hägglund, 2006) as a standard model-free baseline and the Linear Quadratic Regulator (LQR) (Kreindler, 1963), which is optimal control for linear systems under quadratic costs. To handle nonlinear dynamics, we consider switching controllers including Sliding Mode Control (SMC) (Utkin, 2003) and Bang-Bang control; these methods use discontinuous actions to enforce desired behavior but often induce chattering. Additionally, we incorporate an expert heuristic (Energy Shaping)(Spong, 2002), a domain-specific controller explicitly designed to manage the global non-linearity inherent in the Pendulum Swing-Up task.

**Results Analysis.** Table 19 summarizes the results. On Pendulum Swing-Up, where linear controllers typically fail, A$_2$DEPT succeeds and improves upon the energy-shaping heuristic, suggesting a more efficient swing-up strategy. On CartPole and the Double Integrator, A$_2$DEPT performs comparably to the optimal LQR baseline, indicating that it can recover near-optimal feedback behavior in linearizable settings. Moreover, relative to switching baselines that can be destabilized by chattering, A$_2$DEPT tends to produce smooth control laws that remain stable over the long evaluation horizon.

