# OpenReview forum: "$A_2$DEPT: Large Language Model–Driven Automated Algorithm Design via Evolutionary Program Trees"
_ICML.cc/2026/Conference — ICML 2026 regular_

### Official Review · Reviewer_at1u · 2026-03-13

**Soundness:** 3
**Presentation:** 3
**Significance:** 2
**Originality:** 2
**Overall Recommendation:** 4
**Confidence:** 3

**Summary:**

This paper proposes A2DEPT, a framework for LLM-driven Automated Algorithm Design that uses evolutionary program trees to evolve complete solver programs for combinatorial optimization problems. The key idea is to move from template-bound AHD where the LLM fills in a fixed heuristic template to open-ended AAD, where the search target expands from heuristic components to complete solver programs. Experiments on six NP-hard combinatorial optimization benchmarks plus continuous optimization tasks show consistent gains over prior LLM-based AHD methods like FunSearch.

**Compliance With Llm Reviewing Policy:**

Affirmed.

**Final Justification:**

The rebuttal addresses most of my concerns, and I have increased my score.

**Key Questions For Authors:**

1. How does A2DEPT perform when the total token budget not just LLM call count is equalized across all methods?

2. How sensitive is the framework to the choice of LLM?

See weakness for other questions.

**Limitations:**

yes

**Strengths And Weaknesses:**

**Strengths:**

1. **Well-motivated problem framing.** The paper clearly identifies the bottleneck of existing LLM-based AHD: coupling to fixed constructive frameworks limits the design space. The shift from template-bound heuristic generation to open-ended solver synthesis is a natural and important direction.
2. **Comprehensive system design.** The framework is thoughtfully engineered with multiple complementary mechanisms: structured program representation, tree-structured search with hybrid selection, hierarchical operators at three granularity levels, adaptive operator scheduling, diversity-aware partner selection, and a dependency repair loop. Each component is well-motivated.

3. **Broad experimental coverage.** Six NP-hard COP benchmarks across different domains, plus generalization experiments and even continuous optimization. The breadth is impressive and goes beyond what most papers in this area attempt.

**Weaknesses:**

1. The paper is fundamentally a systems paper with many moving parts, but provides no theoretical analysis of why this particular combination should work. There is no convergence analysis, no formal characterization of the search space, and no principled justification for design choices beyond intuition and ablation.

2. While A2DEPT outperforms other LLM-based methods, the optimality gaps on standard benchmarks are still substantial. The the practical value proposition over well-tuned conventional solvers is unclear.

3. The cost-benefit analysis relative to simply running a commercial solver or hand-tuning a known metaheuristic is not discussed.

4. All methods are given the same LLM call budget, but a fairer comparison would normalize by total token budget rather than number of LLM calls.

---

> ### Author Rebuttal · Authors · 2026-03-31
>
> Dear Reviewer at1u,
>
> Thank you for your insightful and valuable comments. We sincerely hope our rebuttal adequately addresses your concerns.
>
> **W1 Theoretical analysis**
>
> > The paper provides no theoretical analysis of why this particular combination should work. There is no convergence analysis, no formal characterization of the search space, and no principled justification for design choices beyond intuition and ablation.
>
> We address design justification and convergence analysis separately.
>
> **1. Principled design rationale.** A2DEPT builds upon a tree-structured search similar to MCTS-AHD, where the core design challenge lies in describing parent–child refinement relationships rather than population-level dynamics. Under this structure, SA-based selection is a natural fit: it satisfies detailed balance and provides a well-characterized exploration–exploitation trade-off. The remaining components form a self-reinforcing closed loop where each creates the precondition for the next: SA's high rejection rate creates the need for Boltzmann recovery → Boltzmann provides stepping stones for macro-mutation breakthroughs → macro-mutation's free-form code necessitates dependency repair → evaluation feedback drives the adaptive scheduler (a multi-armed bandit, Eq. 6). This is not a concatenation of standard techniques—each component is necessitated by the consequences of the previous one. The ablation in Table 4 and the case study in Appendix F.1 confirm that removing any component breaks the loop.
>
> **2. Convergence guarantees.** A2DEPT, like all existing LLM-based AHD methods (FunSearch, EoH, ReEvo, MCTS-AHD), does not provide formal convergence guarantees—this is inherent to evaluation-driven search over an unbounded program space. However, the evolved solvers themselves can embody algorithms with well-understood theoretical properties, e.g., the ILS-based MIS solver with minimum-degree greedy construction (Appendix F.1) has known approximation guarantees, and the evolved ODE solver exhibits implicit-method-like stability (Appendix G).
>
> **W2 & W3 Practical value over conventional solvers**
>
> > The optimality gaps are still substantial. The practical value proposition over well-tuned conventional solvers and the cost-benefit analysis is not discussed.
>
> **1. Automated algorithm design has significant real-world value.** In practice, optimization problems frequently undergo dynamic changes—new constraints, shifting objectives, or new problem variants. Manually redesigning solvers for each change is labor-intensive and requires deep domain expertise. Automated algorithm design enables rapid, on-demand synthesis of solvers that adapt to evolving specifications.
>
> **2. Conventional solvers and AHD/AAD offer complementary strengths.** Well-tuned metaheuristics are the product of decades of research; the gap reflects accumulated human effort, not a fundamental limitation. AHD/AAD excels when specifications change dynamically. Moreover, evolved solvers are interpretable programs that practitioners can inspect and refine using conventional methods.
>
> **3.** This direction has been widely recognized (*Nature*, *ICML*, *NeurIPS*). A2DEPT advances it by extending search from heuristic components to complete solver programs. When real-world constraints shift, AAD synthesizes effective solvers within hours, producing interpretable code that experts can further refine—making it a practical accelerator, not a replacement for conventional methods.
>
> **W4 & Q1 Token budget fairness**
>
> > How does A2DEPT perform when the total token budget, not just LLM call count, is equalized across all methods?
>
> We conducted a token-budget-controlled study using DeepSeek v3.2 (Non-thinking Mode), equalizing total cost at 5 yuan and 10 yuan budgets (details in [Table R1](https://anonymous.4open.science/r/anonymous-rebuttal-5679/rebuttal/tables/Table_R4.png)).
>
> A2DEPT remains the strongest method under both budgets across all four benchmarks, confirming that gains are **not an artifact of consuming more tokens**. When the budget is halved from 10 to 5 yuan, all methods degrade, but A2DEPT preserves the best performance—indicating the advantage comes from the search framework itself rather than larger token consumption. We will include token-budget normalization in the revision.
>
> **Q2 LLM sensitivity**
>
> > How sensitive is the framework to the choice of LLM?
>
> A2DEPT is evaluated under multiple backbones in the main paper (DeepSeek, Gemini, thinking/non-thinking modes) and consistently outperforms baselines. A controlled scaling study on CVRP using Qwen-3.5 (4B–397B, [Table R2](https://anonymous.4open.science/r/anonymous-rebuttal-5679/rebuttal/tables/Table_R3.png)) shows A2DEPT **outperforms all baselines at every scale**, including 4B, with a steeper scaling curve. A2DEPT is not dependent on a single model, but benefits more from stronger models—a strength for exploiting future LLM improvements.

---

> > ### Author Rebuttal · Reviewer_at1u · 2026-04-04
> >
> > I thank the authors for their detailed rebuttal, which has addressed most of my concerns. I will increase my score.

---

> > > ### Author Response · Authors · 2026-04-04
> > >
> > > Thank you very much for reading our rebuttal and for your thoughtful acknowledgement. We are glad that our clarifications were helpful and that they addressed most of your concerns. We greatly appreciate your time and consideration. We are happy that you will increase the score and thanks again.

---

### Official Review · Reviewer_BFYE · 2026-03-13

**Soundness:** 3
**Presentation:** 3
**Significance:** 3
**Originality:** 3
**Overall Recommendation:** 5
**Confidence:** 3

**Summary:**

This paper proposes A2DEPT, a framework that treats LLMs as system-level algorithm architects rather than component-level heuristic generators. Instead of filling slots within a fixed solver template, it evolves complete solver programs through a tree-structured search combining simulated annealing selection, Boltzmann sampling, hierarchical mutation operators, and a program maintenance loop for executability. Experiments on combinatorial optimization benchmarks and ablation studies demonstrated the effectiveness of this framework.

**Compliance With Llm Reviewing Policy:**

Affirmed.

**Final Justification:**

I appreciate the authors for the rebuttal. I will maintain my score.

**Key Questions For Authors:**

- As described in Section 3.2.2, what is the termination condition if the LLM fails to produce a convergent dependency closure within K iterations?
- In L89-90, the paper mentioned “9.8% average gap reduction on our test suite”. How is this number computed? What is the reference baseline to compute reduction here?
- Which LLM backend was used for A2DEPT and LLM baselines in Table 2?

**Limitations:**

yes

**Strengths And Weaknesses:**

### Strengths
- The paradigm shift from template-bound AHD to open-ended AAD is well-motivated.
- The experiments and ablation studies are thorough, demonstrating the effectiveness of the framework.
- The paper is well-organized and easy to follow.

### Weaknesses
- When a child node inherits the updated operator weights from its parent (L256), this assumes that operator preferences that worked well for one program will transfer to its offspring. This is questionable because after a macro-mutation (m₂), the child program may have a completely different algorithmic structure than the parent. The inherited preference for m₁ or m₂ based on the parent's experience is unlikely to be informative for the child's very different optimization landscape. For example, a parent with a high m₂ preference is more likely to create a child with high m₂ preference too, but the child might need m₁ mutation to obtain a high score.
- Because A2Dept depends on many sampling processes (hybrid selection, adaptive operator scheduling), it can be the case that A2Dept does not have reliable convergence (i.e., high deviation). While the paper mentioned reporting results averaged across 3 runs for standard benchmarks and 20 runs for high-constraint benchmarks, there is no included results on standard deviation on them.
- There is a lack of theoretical guarantees for convergence or optimality, which is also acknowledged by the paper in Section 5.

### Suggestions:
- The paper can add another ablation study to verify the effectiveness of the simulated annealing selection by comparing it with random selection.
- The paper can benefit from a coverage chart to show the improvement by number evaluation, as well as a comparison between frameworks for each individual generating.

---

> ### Author Rebuttal · Authors · 2026-03-31
>
> Dear Reviewer BFYE,
>
> Thank you for your insightful and valuable comments. We sincerely hope our rebuttal adequately address your concerns.
>
> **W1 Operator weight inheritance after macro-mutation**
>
> > After a macro-mutation, the child program may have a completely different structure. The inherited operator preference is unlikely to be informative for the child.
>
> The example in the paper is presented for readability and may have been misleading.
> The inherited weights serve as a warm start, not a fixed preference.
> If the inherited bias is unsuitable, Eq. 6 will quickly penalize the underperforming operator and reinforce the effective one within a few iterations. Additionally, the softmax scheduler (Eq. 5) with temperature τ ensures both operators always retain non-zero selection probability, so the less-preferred operator is never excluded before correction occurs.
> We will revise the description in the final version to clarify this mechanism and avoid potential misinterpretation.
>
> **W2 Standard deviations**
>
> > A2DEPT may not have reliable convergence due to many sampling processes. There is no included results on standard deviation.
>
> The complete results with standard deviations are provided in [Table R1](https://anonymous.4open.science/r/anonymous-rebuttal-5679/rebuttal/tables/Table_R1.png) and [Table R2](https://anonymous.4open.science/r/anonymous-rebuttal-5679/rebuttal/tables/Table_R2.png). A2DEPT's variance is comparable to baselines.
>
>
> **W3 & Q1 Convergence guarantees and dependency repair termination**
>
> > There is a lack of theoretical guarantees for convergence or optimality.
>
> We appreciate these questions and address them in turn.
>
> **Theoretical guarantees.** We acknowledge that A2DEPT, like all existing LLM-based AHD methods, does not provide formal convergence or optimality guarantees for the search framework itself.
> This is an inherent property of evaluation-driven program search over an effectively unbounded discrete space, and is not specific to our method.
>
> However, we note that while the search process lacks formal guarantees, the evolved solvers themselves can embody algorithms with well-understood theoretical properties.
> For example, in the MIS case study (Appendix F.1), A2DEPT evolved an ILS-based solver that incorporates greedy construction with minimum-degree selection—a strategy known to provide provable approximation guarantees for MIS on bounded-degree graphs.
> Similarly, in continuous optimization (Appendix G), the evolved ODE solver exhibits implicit-method-like stability properties that are well-characterized in numerical analysis.
> In other words, A2DEPT leverages the LLM's knowledge of theoretically grounded algorithms to synthesize solvers whose quality can be analyzed post hoc, even though the evolutionary search itself is heuristic.
>
> > What is the termination condition if the LLM fails to produce a convergent dependency closure?
>
> **Dependency repair termination.** As specified in Algorithm 1 (Lines 19–22), the repair loop terminates when either (i) all missing dependencies are resolved (U=∅), or (ii) the global LLM call budget is reached (t≥Tmax). If the program remains incomplete after exhausting the budget, it is assigned a score of zero (Lines 25–26) and does not block subsequent search iterations.
>
>
> **S1 & S2 Additional ablation and convergence curve**
>
> > The paper can add an ablation comparing SA selection with random selection, and a convergence chart showing improvement by number of evaluations.
>
> The SA vs. random selection ablation is provided in [Table R3](https://anonymous.4open.science/r/anonymous-rebuttal-5679/rebuttal/tables/Table_R5.png), and the convergence curves are provided in [Figure R1](https://anonymous.4open.science/r/anonymous-rebuttal-5679/rebuttal/figures/convergence.pdf). We will incorporate both in the revised manuscript.
>
>
> **Q2 Computation of 9.8% gap reduction**
>
> > How is this number computed? What is the reference baseline to compute reduction here?
>
> The reported 9.8% average gap reduction is computed over normalized, comparable quantities. Specifically, the gap metric for each problem is defined as the relative percentage deviation from the optimal solution.
> Since all values are already normalized to the same percentage scale before averaging, the cross-problem mean is not dominated by any single problem's raw objective magnitude. That said, we acknowledge that a single aggregate number may obscure per-problem variation, and we will revise the presentation in the final version to make the per-problem breakdown more prominent alongside the average, ensuring full transparency.
>
>
> **Q3 LLM backend for Table 2**
>
> > Which LLM backend was used for A2DEPT and LLM baselines in Table 2?
>
> All methods in Table 2 use DeepSeek v3.2 (Non-Think Mode), consistent with the first group in Table 1.
> We will explicitly annotate this in the revised manuscript to avoid ambiguity.

---

> > ### Author Rebuttal · Reviewer_BFYE · 2026-04-04
> >
> > Thank you for your detailed response. All my concerns are addressed. I will keep my original score.

---

> > > ### Author Response · Authors · 2026-04-04
> > >
> > > Thank you very much for reading our rebuttal and for your thoughtful acknowledgement. We appreciate your time and consideration.

---

### Official Review · Reviewer_ea7C · 2026-03-13

**Soundness:** 3
**Presentation:** 3
**Significance:** 2
**Originality:** 2
**Overall Recommendation:** 4
**Confidence:** 4

**Summary:**

This paper introduces an new automated algorithm design framework, which treating LLMs as system-level architects and coding agent. The framework performs tree-structured evolutionary search to enable structured exploration and reuse of historical feedback.

**Compliance With Llm Reviewing Policy:**

Affirmed.

**Key Questions For Authors:**

For industrial large-scale algorithms, such as writing or optimizing mathematical programming solver may involve dozens of files and hundreds of thousands of lines of code, is the tree-search framework encounter inherent scalability limitations as the codebase grows?

**Limitations:**

yes

**Strengths And Weaknesses:**

Strengths: Its core strength lies in the technical innovation of using a tree-structured evolutionary search with a robust maintenance loop to ensure code executability. The Soundness and Presentation is not bad.
Weakness: the paper faces challenges regarding the reliability of long-horizon synthesis and the potential scalability bottlenecks of its reactive validation pipeline. Meanwhile, the concepts of open-ended, system-level algorithm design and global search trees are not entirely novel.

---

> ### Author Rebuttal · Authors · 2026-03-31
>
> Dear Reviewer ea7C,
>
> Thank you for your insightful and valuable comments. We sincerely hope our rebuttal adequately address your concerns. If so, we would deeply appreciate it if you could consider raising your score. If not, please let us know your further concerns, and we will continue actively responding to your comments.
>
> **W1.1 & Q1 Reliability of long-horizon synthesis and scalability**
>
> > The paper faces challenges regarding the reliability of long-horizon synthesis and the potential scalability bottlenecks of its reactive validation pipeline. Is the tree-search framework encounter inherent scalability limitations as the codebase grows?
>
> We thank the reviewer for raising this important question. We address reliability and scalability separately below.
>
> **1. Reliability is maintained through the program maintenance pipeline.** As discussed in Section 3.2, A2DEPT enforces executability through three complementary mechanisms: structured program representation with role-aware parsing confines edits to mutable strategy code; dependency repair resolves missing references via call-graph analysis; and reachability pruning removes dead logic after each edit.
> Additionally, the overhead of the validation pipeline is moderate: Table 11 shows that A2DEPT's wall-clock time (2.5–3.0h) and token consumption (0.6–0.7M input) remain comparable to or lower than MCTS-AHD (3.0–3.5h, 1.0–1.4M input), indicating that the repair loop does not introduce a scalability bottleneck within the current problem scope.
>
> **2. Current scope and industrial-scale limitations.** We acknowledge that A2DEPT is designed for algorithmic-level solver synthesis, where evolved programs typically comprise hundreds of lines of modular code.
> However, we agree that scaling to industrial-level codebases (hundreds of thousands of lines across dozens of files) would require architectural extensions beyond the current design—such as hierarchical module management, incremental compilation, and file-level dependency tracking.
> We view A2DEPT as establishing the foundational paradigm for program-level solver evolution; extending it to larger software systems is an important direction for future work.
>
> ---
>
> **W1.2 Novelty of open-ended AAD and global search tree**
>
> > The concepts of open-ended, system-level algorithm design and global search trees are not entirely novel.
>
> We appreciate this comment. We agree that open-ended program synthesis and tree-structured search are not individually new concepts.
> We clarify below the specific contributions of our work from the paradigm level down to empirical validation.
>
> **1. AAD offers a fundamentally broader design space than AHD.** As demonstrated in Figure 1(a), heuristics evolved under one template fail to transfer to another, and the relative advantage of framework choices is problem-dependent.
> This means template-bound AHD inherits a hidden design decision—selecting the solver backbone—that can cap performance regardless of heuristic quality. AAD removes this ceiling by making the full algorithmic workflow evolvable.
> This paradigm shift is not merely conceptual: Table 2 shows that simply moving existing methods (EoH, ReEvo) from AHD to AAD does not close the gap to A2DEPT, confirming that open-ended freedom alone is insufficient without proper mechanisms to manage it.
>
> **2. A2DEPT makes open-ended exploration practical through bounded search.**
> The critical challenge of AAD is not the concept itself, but making it work reliably.
> A2DEPT addresses this through structured program representation, dependency repair, and hierarchical operators that collectively keep the non-executable rate at a manageable level, enabling sustained long-horizon evolution.
>
> **3. Concrete advances beyond existing work.** Compared to MCTS-AHD, which applies tree search to component-level heuristic design within fixed templates, A2DEPT extends the search target to complete solver programs and introduces the program maintenance loop to handle the resulting executability challenges.
> Compared to EoH and ReEvo operating in AAD mode (Table 2), A2DEPT's tree-structured hybrid selection and adaptive scheduling preserve long-range logical consistency across edits—a capability absent in population-based methods that treat each program independently.
>
> **4. Broad empirical validation from standard to out-of-domain settings.** A2DEPT's advantages are validated across progressively challenging scenarios:
> (i) standard CO benchmarks covering graph, routing, location, and scheduling domains (Table 1);
> (ii) highly constrained problems with fragmented feasible regions (CEVRPTW, MRCPSP, Table 3);
> (iii) continuous optimization tasks (Tables 12–14), demonstrating that the framework generalizes beyond discrete combinatorial optimization.

---

> > ### Author Rebuttal · Reviewer_ea7C · 2026-04-03
> >
> > Thanks for the authors' response, after reading the rebuttal, I think it is appropriate to maintain my current score.

---

> > > ### Author Response · Authors · 2026-04-04
> > >
> > > Dear Reviewer ea7C,
> > >
> > > Thank you very much for reading our rebuttal and for your thoughtful acknowledgement. We appreciate your time and consideration.

---

### Official Review · Reviewer_vtxy · 2026-03-13

**Soundness:** 3
**Presentation:** 3
**Significance:** 2
**Originality:** 2
**Overall Recommendation:** 4
**Confidence:** 3

**Summary:**

This paper proposes a framework that shifts LLM-based optimization from template-bound Automated Heuristic Design to open-ended Automated Algorithm Design, where LLMs evolve complete solver programs rather than isolated heuristic components. The key mechanisms proposed by this paper include a tree-structured evolutionary search with hybrid selection (SA + Boltzmann), hierarchical operators (micro-tuning, macro-mutation, semantic crossover) with adaptive scheduling, and a program-maintenance loop for dependency repair and executability enforcement. Experiments on six NP-hard CO benchmarks show consistent improvements over AHD baselines.

**Compliance With Llm Reviewing Policy:**

Affirmed.

**Final Justification:**

The paper is well-motivated in shifting from template-bound AHD to open-ended solver synthesis, with solid ablation and insightful case studies. My main concerns about the heterogeneous gap metrics, time-limit fairness, component novelty, and LLM dependency were all convincingly addressed in the rebuttal. Originality remains moderate as individual components are standard, but the closed-loop argument for their integration is reasonable. Overall I will keep my positive rating.

**Key Questions For Authors:**

1. Could you provide standard deviations for Tables 1 and 2 across the 3 runs? In my experience, the LLM-based generation is generally unstable.

2. Table 2 shows GLS-based MCTS-AHD achieves 10.76% gap on FJSP versus the proposed model’s 18.95%. Could the authors characterize more precisely what structural properties of a problem make template-bound AHD with domain-specific operators preferable to open-ended AAD?

3. The evolved solvers are evaluated on the same benchmark distributions used during evolution. Have the authors tested whether the best evolved solver generalizes to out-of-distribution instances?

**Limitations:**

Yes

**Strengths And Weaknesses:**

**Strengthes**

1. This paper is well-motivated, with a move from heuristic-slot optimization to full solver synthesis.

2. The case study figure 3 and figure 4 is valuable. It concretely shows how the system transitions from a greedy heuristic to an ILS-style solver, and honestly analyzes failure modes (bitmask refactoring trap).

3. The ablation study design is solid, systematically removing each component (fixed-template, no Boltzmann, no adaptive scheduling), and shows consistent degradation.


**Weaknesses**

1. The authors claim that “reducing the optimality gap by an average of 9.8%”, but this average gap reduction is computed over heterogeneous problems with very different gap magnitudes. On FJSP, the proposed model still lags behind GLS-based AHD methods. I would suggest transparently discussing these.

2. The evaluation uses a 120-second wall-clock limit per solver run on the test set. But the evolved solvers vary dramatically in complexity (greedy vs. ILS with local search). It is unclear whether the time limit systematically advantages or disadvantages certain algorithmic paradigms. It might be good to have a Pareto analysis between runtime vs. quality.

3. Several individual components are standard techniques, like the SA-based selection, Boltzmann sampling, adaptive operator scheduling, and dependency graph repair. I feel that the novelty lies in their combination for LLM-driven program evolution, but the paper could better articulate why this specific assembly is more than the sum of its parts beyond the ablation.

4. The LLM sensitivity study shows that the proposed method degrades more than baselines when switching to a weaker model (Gemini Flash-lite). This is acknowledged but not addressed. The proposed method's gains may be partly attributable to the strong reasoning model rather than the framework itself.

---

> ### Author Rebuttal · Authors · 2026-03-31
>
> Dear Reviewer vtxy,
>
> Thank you for your kind support and valuable comments. We sincerely hope our rebuttal adequately addresses your concerns.
>
> **W1.1 Heterogeneous gap reduction metric**
>
> > The average 9.8% gap reduction is computed over heterogeneous problems with very different gap magnitudes.
>
> The 9.8% is computed over normalized Gap (%) values—relative percentage deviations from optimal—so all problems are on the same scale before averaging. We will make the per-problem breakdown more prominent in the revised manuscript.
>
> **W1.2 & Q2 FJSP performance & structural analysis**
>
> > On FJSP, the proposed model still lags behind GLS-based AHD methods. What structural properties make template-bound AHD preferable to open-ended AAD?
>
> The GLS advantage on FJSP stems from its domain-specific operators, not LLM-designed knowledge. We compared GLS with an LLM-designed matrix ($GLS_{LLM}$) and a random matrix ($GLS_{rand}$):
>
> Table R1: Normalized scores under different GLS.
>
> |    | $GLS_{LLM}$ |$GLS_{rand}$|
> |----|-------------|------------|
> |CFLP| 0.621       | 0.302      |
> |CVRP| 0.845       | 0.830      |
> |FJSP| 0.873       | 0.808      |
> | MIS| 0.844       | 0.818      |
>
> On FJSP, random guidance still achieves 0.808, indicating strong inductive bias from the fixed neighborhood operators. The much larger drop on CFLP shows that LLM-designed knowledge matters when operators alone are insufficient. Thus, A2DEPT and template-bound AHD are complementary.
>
> **W2 Wall-clock time limit fairness**
>
> > The 120-second wall-clock limit may systematically advantage or disadvantage certain algorithmic paradigms.
>
> Table R2: Normalized scores under six time budgets.
>
> | Time (s) | 30    | 60    | 120   | 180   | 240   | 300   |
> |----------|-------|-------|-------|-------|-------|-------|
> | A2DEPT   | 0.905 | 0.907 | 0.914 | 0.919 | 0.928 | 0.925 |
> | EoH      | 0.792 | 0.811 | 0.819 | 0.824 | 0.823 | 0.823 |
>
> Both methods improve as the budget increases. The interval between 120s and 180s appears to be a particularly reasonable operating range. We therefore view the 120s setting used in the paper as a practical and balanced choice.
>
> **W3 Novelty of component assembly**
>
> > Individual components are standard. The paper should better articulate why this assembly is more than the sum of its parts.
>
> A2DEPT builds upon a tree-structured search where the core challenge lies in describing parent–child refinement relationships rather than population-level dynamics, and SA-based selection is a natural fit for this setting. The key insight is that these components are not independent modules—they form a self-reinforcing closed loop where each creates the precondition for the next:
>
> - SA's high rejection rate under open-ended AAD creates the need for Boltzmann sampling to recover promising but temporarily underperforming branches.
> - The diversity preserved by Boltzmann provides stepping stones for macro-mutation to achieve paradigm-level breakthroughs.
> - Macro-mutation's free-form code necessitates dependency repair to maintain executability.
> - Evaluation feedback drives the adaptive scheduler to learn which operator granularity is productive, shaping the next cycle.
>
> **W4 LLM sensitivity & model dependency**
>
> > Gains may be partly attributable to the strong reasoning model rather than the framework itself.
>
> A controlled Qwen-3.5 scaling study (4B–397B) on CVRP shows that A2DEPT outperforms all baselines at every scale (details in [Table R3](https://anonymous.4open.science/r/anonymous-rebuttal-5679/rebuttal/tables/Table_R3.png)). Its steeper scaling trend suggests that A2DEPT more effectively converts LLM capability into optimization performance. At the same time, the framework does require a minimum level of code-generation ability, which explains the degradation observed with Gemini 2.5 Flash-lite.
> **Q1 Standard deviations**
>
> > Could you provide standard deviations for Tables 1 and 2?
>
> The standard deviations are provided in [Table R4](https://anonymous.4open.science/r/anonymous-rebuttal-5679/rebuttal/tables/Table_R1.png) and [Table R5](https://anonymous.4open.science/r/anonymous-rebuttal-5679/rebuttal/tables/Table_R2.png).
>
>
> **Q3 Out-of-distribution generalization**
>
> > Have the authors tested whether the best evolved solver generalizes to out-of-distribution instances?
>
> We tested on VRPTW Solomon instances with three spatial distributions, evolving solvers on each and testing across all:
>
> Table R6: Mean Gap (%) of evolved solvers tested across distributions on VRPTW (Solomon benchmark).
>
> | Evolve\Test | C     | R     | RC    |
> |-------------|-------|-------|-------|
> | C           | 23.05 | 32.52 | 28.77 |
> | R           | 24.37 | 28.40 | 26.07 |
> | RC          | 25.87 | 31.07 | 28.50 |
>
> Evolved solvers do not collapse on OOD distributions—performance degrades gracefully. The R-trained solver achieves the best overall performance, confirming A2DEPT discovers generalizable strategies rather than overfitting to distribution-specific patterns.

---

> > ### Author Rebuttal · Reviewer_vtxy · 2026-04-04
> >
> > Thanks for the thorough rebuttal.
> >
> > The key concerns are well addressed: the normalized gap clarification, the GLS random-matrix experiment nicely showing complementarity with template-bound AHD, the time-budget sweep supporting the 120s choice, and the OOD generalization results. The scaling study across Qwen model sizes also partially mitigates my LLM dependency concern.
> >
> > I will maintain my positive score.

---

> > > ### Author Response · Authors · 2026-04-04
> > >
> > > Thank you very much for reading our rebuttal and for your thoughtful acknowledgement. We appreciate your time and consideration.

---

### Decision · Program_Chairs · 2026-04-30

**Decision:**

Accept (regular)

**Comment:**

The paper introduces an LLM-driven framework that synthesizes programs for combinatorial optimization, using a tree-structured evolutionary search with hybrid selection and hierarchical operators. The problem is well motivated (improving Automated Heuristic Design) and the proposed solution is a practical synthesis of established techniques into a coherent design with significant performance benefits. The exposition is clear and the reviewers agree that the experiments and ablations are thorough, though formal theoretical guarantees are lacking. During the rebuttal phase, the authors addressed key concerns by providing per-problem breakdowns, running time-budget and token-budget experiments, studying sensitivity to LLMs and OOD generalization. The reviewers all acknowledged that their main issues were resolved post-rebuttal. Incorporating the changes introduced during the rebuttal will substantially strengthen the subsequent version of this paper.